# COOPERA: Continual Open-Ended Human-Robot Assistance

**Chenyang Ma**[1]     **Kai Lu**[1]
**Ruta Desai**[†]   **Xavier Puig**[†]   **Andrew Markham**[1†]   **Niki Trigoni**[1†]
[1]University of Oxford

## Abstract

To understand and collaborate with humans, robots must account for individual human traits, habits, and activities over time. However, most robotic assistants lack these abilities, as they primarily focus on predefined tasks in structured environments and lack a human model to learn from. This work introduces **COOPERA**, a novel framework for COntinual, OPen-Ended human-Robot Assistance, where simulated humans, driven by psychological traits and long-term intentions, interact with robots in complex environments. By integrating continuous human feedback, our framework, for the first time, enables the study of long-term, open-ended human-robot collaboration (HRC) in different collaborative tasks across various time-scales. Within COOPERA, we introduce a benchmark and an approach to personalize the robot's collaborative actions by learning human traits and context-dependent intents. Experiments validate the extent to which our simulated humans reflect realistic human behaviors and demonstrate the value of inferring and personalizing to human intents for open-ended and long-term HRC.

## 1   Introduction

A long-standing goal in robotics is to develop agents that can effectively assist humans in their daily lives by adapting to their preferences and habits. In order to do this, a robot agent must be able to not only learn to interact in environments with humans in a given moment, but also reason about the human across long periods of time, adapting its behavior to provide better assistance. For example, such an agent should be able to fetch a cup of coffee while also understanding that someone may prefer it cooler in the morning but stronger in the afternoon, heating up water accordingly.

Over recent years, several works have made significant advances in developing agents that can assist humans in household tasks [52, 50, 75, 65, 47], using simulation environments to study human-robot collaboration (HRC) in a safe and scalable manner. However, most of these works focus on episodic settings, where a robot is evaluated over a set of short collaboration scenarios with tasks specified in advance. These settings are very different from real-world scenarios, where humans have preferences and long-term goals that guide their behaviors, needing different types of assistance at different times.

To advance robot agents that can assist and adapt to humans, we propose **COOPERA**, a novel framework for COntinual, OPen-Ended human-Robot Assistance in complex household environments (Fig. 1). At its core, COOPERA features a human model with preferences that supports long-term interactions, a feedback mechanism, and benchmarks and metrics to evaluate if robots can assist humans in long-term tasks and reason about their preferences effectively.

To model realistic humans, we simulate humans using an LLM with detailed human traits and habits, retrieved environment information, and intention history, enabling behaviors that exhibit three characteristics. **1) Dynamic intention-driven:** Humans act based on intentions that vary over time

---
[†]Equal advising.

39th Conference on Neural Information Processing Systems (NeurIPS 2025).

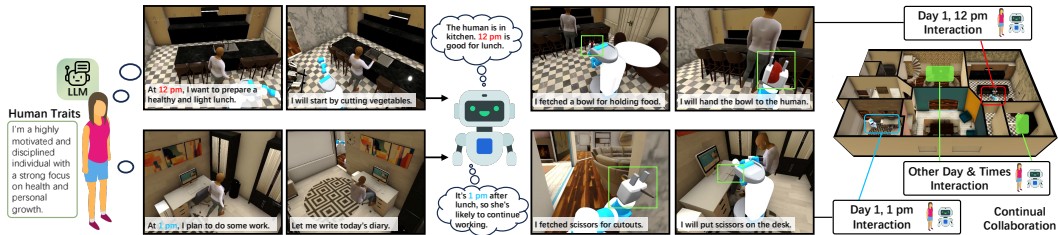

Figure 1: **Continual human-robot collaboration for open-ended tasks over multiple days.** Our framework **COOPERA** entails an approach to simulate traits-driven humans with long-term, whole-day behaviors within robot simulation platform, enabling the first study of long-term, open-ended human-robot collaboration. We also introduce a benchmark and a method for the robot to personalize collaboration in such continual, open-ended settings by learning human traits and context-dependent intents over time.

(e.g., setting the dinner table at 6 pm, then watching TV at 7 pm). **2) Open-ended and environment-conditioned:** Rather than following predefined tasks, humans generate spontaneous intentions based on the environment, available objects, and time of day. **3) Traits-driven:** Psychological traits and habits shape human behavior, resulting in diverse routines even within similar environments (e.g., one person starts their day reading, while another prefers cleaning).

As the human interacts in the environment, we need a way to provide feedback to the robot so it can improve over time. We structure our framework into two stages which happen on each day of interaction. At the beginning of the day, the robot observes and collaborates with the human, assisting in inferred tasks. At the day's end, the human communicates with the robot and provides feedback to help improve the robot's collaboration success rate for subsequent days.

COOPERA presents unique challenges that are often overlooked in existing HRC benchmarks. First, robot agents need to reason not only about the environment state but also a given human's behavior for effective assistance. Rather than learning a behavior that assists all possible humans, they need to adapt to each person's preferences and traits. Second, humans in our framework perform different tasks depending on the time of the day or the day of the week (e.g., they may only do exercise once). Thus, effective agents need to reason about time as a cue for how to best assist the human.

To explore this challenging framework, we provide a benchmark and propose a method that tracks human preference profiles and uses VLMs and classifiers to predict and score human goals based on time, observed behavior, and profile, suggesting actions to achieve these intentions. This enables the robot to learn and mimic human behavior by capturing the underlying correlations between human traits, temporal dependencies, and their corresponding intentions and tasks. We compare our method against several baselines, evaluating the robot's collaboration performance over multiple days in different collaborative tasks and across diverse humans and scenes. Furthermore, we conduct extensive experiments to assess how well our simulated humans reflect real human behavior, particularly their ability to exhibit distinct, trait-driven patterns aligned with human profiles.

In summary, our main contributions are threefold:

- We present COOPERA, a novel HRC framework for continual, open-ended collaboration with humans who exhibit individual traits across long horizons.
- We develop a method to simulate humans with long-term behavior models driven by individual traits and habits.
- Within this framework, we introduce a benchmark and an approach that enables increasingly adaptive and personalized collaboration with humans over multiple days.

## 2 Related Work

**Human-Robot Collaboration.** Prior HRC work has largely focused on controlled lab settings [19, 10, 44, 59], where collaborative tasks are shared by both the human and the robot or narrowly defined. More recent research has expanded to complex household environments, requiring robots to infer human intentions from a single demonstration [50, 20, 64] or in an online fashion [51]. Subsequent works explore human intention inference using data from images [39] or simplified environments (e.g., 2D worlds) [5, 55, 70, 68], progressing to simulated real-world environments [20] and leveraging recent advances in VLMs. However, these approaches typically rely on predefined, closed-form representations of human intentions and tasks [14, 32, 2, 12, 71], and often ignore realistic human

behavior. Furthermore, collaboration is usually limited to fixed episodes with predefined task set. In contrast, our work considers open-ended and continual HRC, where humans spontaneously propose their actions based on environmental factors, and the collaboration persists across days.

**Human Simulation.**  Most embodied AI works [6, 3, 16, 17, 60] assume that environmental changes are solely driven by a single robot [52]. Due to the challenges of real-human experiments (e.g., safety, scalability, cost), recent research has integrated deformable humans with plausible motion and appearance into robot simulation platforms [49, 52], enabling the study of safe and scalable HRC. However, these simulated humans focus only on motion feasibility, lacking the complexity and variability of real human behavior. Another research direction simulates humans with psychological traits and social interactions [76, 43, 46, 26, 64], but remains language-based and does not involve environmental interaction. In contrast, we simulate humans driven by psychological traits and habits, whose behavior is long-term and capable of interacting with their environment.

**LLMs for Human Task Inference.**  One line of research treats human intentions as direct inputs and investigates how LLMs can interpret open-ended natural language instructions to generate structured robot plans [25, 24, 23, 73, 63, 38]. These works use techniques such as 3D scene graphs [56, 36, 8] to semantically ground high-level goals and decompose them into actionable subgoals, or incorporate human feedback [58, 33, 9] to quantify uncertainty and enable skill acquisition through interaction. In contrast, COOPERA takes a step further by aiming to let LLMs/VLMs infer personalized task plans, adapting to specific human traits and habits rather than general commonsense knowledge.

## 3  COOPERA: Continual, Open-Ended HRC Framework

Our goal is to enable the study of continual HRC in open-ended tasks. To that end, we investigate how a robotic agent can become more effective in assisting humans by learning from their behavior. Central to COOPERA are LLM-powered simulated humans driven by traits and long-term intentions that the robot can reason for effective collaboration, and a human feedback mechanism for improving collaboration over time. We first outline our framework and problem setup, detailing the collaboration settings we explore (Fig. 2). Then, we describe our approach of simulating humans driven by traits with long-term behaviors (Fig. 3). Finally, we propose a method to tackle our framework (Fig. 4).

### 3.1  Overview

In order to investigate HRC in a safe and reproducible manner, we consider a simulated human agent that interacts in a 3D household environment to achieve a set of high-level goals. These goals vary throughout the day and are driven by human traits and habits, as well as by the activities the human has done before. The robot's goal is to assist the human in those tasks, without receiving explicit commands about the goal they should help with, or information about the human's traits. Both human and robot have full knowledge of the environment. Each day is represented as 12 one-hour intervals covering the time from 9 am to 9 pm (the rest is treated as being asleep). At the beginning of each hour, the human proposes a high-level intention (e.g., leisure) and decomposes it into a sequence of tasks and executes them in the environment (e.g., watch TV on the sofa). As the human interacts in the environment, the robot has to infer the

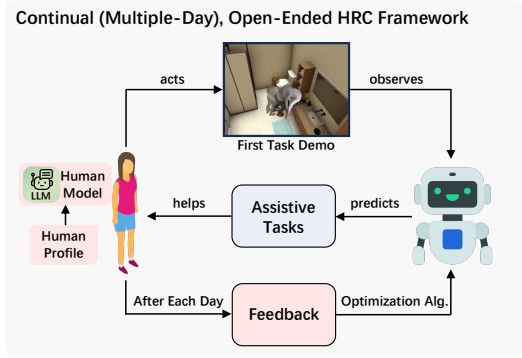

Figure 2: **COOPERA: Continual, open-ended human-robot collaboration framework.** The LLM-powered human proposes whole-day intentions and tasks, executed in the environment. As the robot observes the human actions, it predicts a set of tasks to assist them. After each day, the human provides feedback to the robot, enabling the robot to improve for subsequent days.

human's goals and provide assistance. At the end of each day, the human provides feedback on the robot's help, which is then used to improve the robot's collaboration success in subsequent days.

**Problem Setup.**  We define two types of collaboration with increasing difficulty and openness. **Collaboration type 1** is an open-ended variant of the Watch-and-Help challenge [50], where one intention (e.g., set up dinner table) is decomposed into 3 pick-and-place tasks (i.e., picking an object and placing it on a static object). For each intention, the robot is given a video of the human

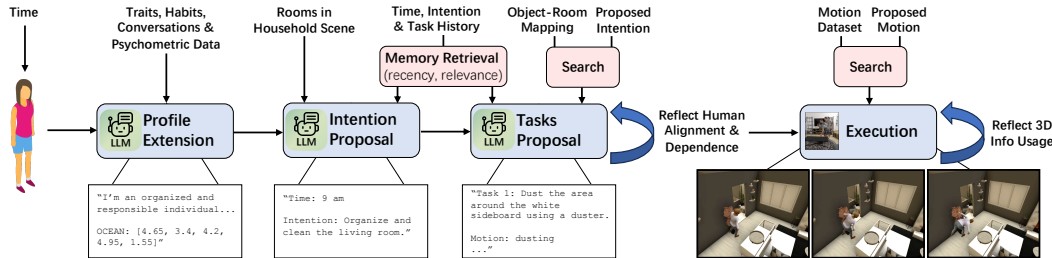

Figure 3: **Human Simulation Pipeline.** We seed the human-LLM with an extended profile. At each time of day, the human proposes an intention and decomposes it into tasks, aligning with profile traits and temporal dependence on intention/task history. LLM inputs are optimized with *Memory Retrieval* and *Search*, and robustness is enhanced via two rounds of *Reflexion*. This pipeline generates continuous, whole-day intentions and tasks executed in the environment with expressive whole-body motion. See Appendices C and F for details.

performing the first task and its textual description, and must infer and assist with the remaining tasks based on objects available in the scene. **Collaboration type 2** is more challenging and moves beyond pick-and-place: each intention (e.g., morning hygiene) is decomposed into 5 tasks involving free-form human motion while interacting with static objects (e.g., the human brushes teeth at the mirror). Unlike type 1, the robot is unconstrained by the scene and may propose any object it deems helpful (e.g., the robot offers toothbrush). It receives only the first task video with no textual guidance.

**Evaluation Settings.** We define four progressively challenging settings. **1) Same human, same scene:** The robot collaborates with the same human in the same scene over 5 consecutive days (5 days, 1 scene). **2) Same human, different scenes:** The robot collaborates with the same human across 5 different scenes, with a new scene each day (5 days, 5 scenes). **3) Different humans, same scene:** The robot collaborates with different humans in the same scene, rotating among Human 1, 2, and 3, each for one day, repeating this cycle three times in the same scene (9 days, 1 scene). **4) Different humans, different scenes:** The robot collaborates with different humans across multiple scenes, rotating through Human 1, 2, and 3 in the first scene, then repeating this sequence in the second and third scenes (9 days, 3 scenes). In **3)** & **4)**, we explore if knowledge gained from interacting with different humans improves future collaboration, despite fewer interaction days per human.

### 3.2 Simulating Humans

We aim to model humans who interact in the environment over long periods of time, act driven by their goals, preferences, and context, and who can react and provide feedback as a robot assists them. To achieve this, we propose a hierarchical model that combines LLMs and 3D human motion to simulate long-term, realistic human behaviors in indoor environments. First, our model generates a description of human traits describing their preferences and habits. Based on these traits, the environment, and the history of human actions, the model then generates a sequence of tasks for the human to perform. For every task, we use the environment information to generate human motions and interactions, providing a realistic demonstration of each task. Fig. 3 shows an overview of our design. Next, we describe in more detail each of the human simulation components.

**Generating Human Traits.** We use LLMs to generate personality traits that determine the human long-term behaviors. For this, we sample conversations from the Synthetic Human Dataset [26], containing dialogues between different humans, and prompt an LLM to generate a description of the human based on the conversation, inferring attributes such as their job, preferences or common activities. Inspired by [67, 77, 48], we also prompt the LLM to generate a vector measuring Big-5 human personality traits [18] (openness, conscientiousness, extroversion, agreeableness, and neuroticism), allowing us to measure the diversity across generated humans and how well the robot agents can infer the human's personality from their interactions.

**Whole-Day Intentions and Tasks.** Given human traits, we generate long-term human behaviors, with tasks featuring temporal dependences within a day and diversity across days. *Temporal Dependence:* Given 3D environment information, we use an LLM to propose intentions for different times of day (e.g., 9 am: clean the living room). Next, we prompt the LLM to decompose the intentions into a sequence of inter-dependent tasks (e.g., dust the area around the white board, clean the counter). The LLM also receives the human's intention and task history from previous hours, and is explicitly prompted to consider their inter-dependency. *Varying Distribution:* While humans with specific traits follow general routines, their daily behavior varies daily (i.e., Monday 9 am for cleaning, Tuesday 9

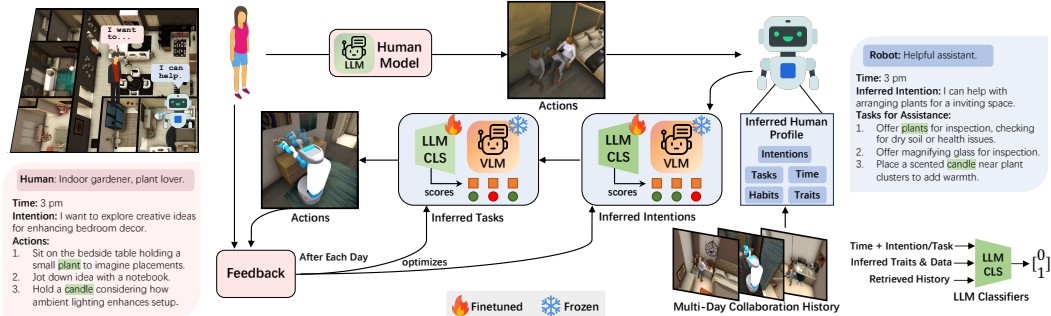

Figure 4: **Our approach for human assistance.** We decouple robot task inference into intention and task inference. By chaining VLM and classifier, the robot selects tasks aligned with the human's traits and temporal context. It maintains a human profile inferred from collaboration history, which, combined with feedback, optimizes the robot-VLM via prompting and the classifiers via supervised learning. See Appendices D and G for details.

am for exercise). To model this, we reset the intention and task history at the start of each new day, setting a high temperature for the human-LLM to encourage diversity across days.

**Expressive Whole-Body Motion.** We simulate human agents physically using expressive 3D whole-body motions during task execution [31, 21, 42] by chaining motion sequences for each task. For pick-and-place tasks, the sequence includes walking, reaching and picking, walking again, then reaching and placing. For tasks involving free-form motion (e.g., sitting on a sofa), the human-LLM describes a free-form human motion that matches each task, using examples from our human motion dataset. The resulting sequence combines walking with the selected free-form motion.

**Optimizing Long-Context Inputs.** Our progressive prompting chain provides the human-LLM with substantial information at each stage, especially during the task proposal stage, where the 3D environment may contain hundreds of objects and the motion dataset includes thousands of data points. Additionally, as the day progresses, the intention and task history grows long (e.g., from 9 am to 9 pm, 13 intention sentences and dozens of task descriptions accumulate). Since LLMs struggle with long-context inputs [30, 35, 69], we introduce *Search* and *Memory Retrieval* mechanisms. *Search:* Given a query text and a list of texts, we return the top-K most relevant items based on semantic similarity. *Memory Retrieval:* We use recency and relevance scores to retrieve the top-K memories. Recency decays over time with a decay factor $\lambda$ from the current time, and relevance is calculated by semantic similarity, similar to search. The final retrieval score is the product of both [46].

**Self-Corrections.** Given the complexity of our progressive prompting process and the LLM responses, even state-of-the-art models can make mistakes. Therefore, during the most complex task proposal stage, we perform two rounds of *Reflexion* [61, 74, 72] to identify and correct errors related to human traits, temporal dependencies, and object use within the 3D environment.

### 3.3 Instantiating COOPERA with an Assistive Agent

To study COOPERA, we propose an approach (Fig. 4) for continual HRC, enabling the robot to learn correlations between human intentions, tasks, traits, and temporal dependencies at each time of day.

At any given time, a human's intentions/tasks can be viewed as meta-intentions/meta-tasks, encompassing a range of possible options due to the diversity of human behavior across days. Our solution decouples task inference into two stages: first inferring intentions, then identifying specific tasks. We capture the correct sets by chaining VLM to imagine multiple possible intentions/tasks and classifiers to score and filter them. Given observation (frames uniformly extracted from a video $V = [f_1, \ldots, f_N]$) of the human's first task, the robot-VLM generates an intention superset. For each positively classified intention by the intention classifier, the robot-VLM infers a set of possible tasks, forming a task superset. The task classifier then identifies the tasks most suitable for collaboration.

We optimize the robot-VLM through prompting and the binary classifiers via supervised learning. Using human feedback from the end-of-day discussion, the robot keeps tracks of a human profile by prompting robot-VLM to infer and summarize the human's traits, habits, and psychometric data. This human profile, along with the retrieved history of intentions and tasks, is incorporated into the robot-VLM prompts and provided as input to the classifiers in the subsequent times and days. The robot-VLM and classifiers are optimized per day. Please see Fig. 4 for the input data format.

Table 1: **Evaluation of 1)** human classification, **2)** simulated human diversity, **3)** human traits-psychometrics coherence, **4)** temporal dependence, and **5)** user studies.

| Classification (Acc ↑) | | Diversity (SD ↑) | Coherence (R ↑) | | Temporal Dependence | | User Studies (Acc ↑) | |
|---|---|---|---|---|---|---|---|---|
| intention | task | | aligned | mismatched | Acc ↑ | F1 ↑ | MCQ | Matching |
| 0.995 | 0.830 | 0.939 | 0.342 | -0.497 | 0.789 | 0.790 | 0.764 | 0.712 |

## 4 Experiments and Analysis

Within COOPERA, we first examine 1) if the central component, the simulated human model, reflects real human behavior and to what extent. 2) We then introduce the benchmark setup (baselines, evaluation metrics) and explore if our proposed approach leads to more personalized robot assistance over multiple days compared to baselines. 3) Subsequently, we analyze the real-world applicability of our framework. 4) Finally, we evaluate the effectiveness of each module through ablation studies.

### 4.1 Framework Implementation

**Environment and Scene.** We use *Habitat 3.0* [52] as the robot simulation platform and *HSSD* [28] as the 3D environment, which includes 18,656 static objects across diverse scenes in style and size. Since the original HSSD includes only static objects, we develop a systematic approach to create dynamic scenes by making small objects from specific categories (e.g., decor, kitchenware) movable. We also sample 20 dynamic objects from the *YCB Dataset* [7] and place them in contextually appropriate locations (e.g., a mug on a bedside table) using Habitat's built-in tools. Dynamic scenes are initialized at the start of each episode. Across days, Habitat tracks object locations as the human and robot interact with the environment, allowing them to maintain updated environment knowledge. We select 5 scenes with varying of rooms $(4-11)$, static objects $(51-140)$, and dynamic objects $(33-94)$. All scenes provide enough space for the human and Fetch robot [15] to navigate. Please see Appendix B for more details.

**Human Dataset.** For modeling unique humans, we use the *SPC: Synthetic-Persona-Chat Dataset* [26], a fully synthetic dataset that includes hundreds of short user profiles along with their conversations and compute psychometric data (details in Appendix C). We use *Motion-X* [31] and *AMASS* [41] as the human motion dataset. We generate 10 human profiles.

**Training and Inference.** We use open-source models for interpretability and benchmarking value. For simulating humans, we use Llama-3.1-8B [13] with temperature 0.7. For search and memory retrieval, we use MiniLM-L6-v2 [66] with a decay factor $\lambda = 0.95$, retrieving the top 3 intentions and top 5 tasks. For the assistive agent, we use Llama-3.2-11B [13] as the robot-VLM. Classifiers are finetuned on Mistral-7B-Instruct-v0.2 [27] using LoRA [22] in instructional format to output binary yes/no. We train on 3 NVIDIA A10 GPUs (24GB RAM). Please see Appendix D for more details.

### 4.2 Analysis of Human Simulation

**Distinct Simulated Humans.** We examine if simulated humans with distinct Big-5 traits exhibit machine-identifiable features. Each of 10 humans is placed in 5 scenes, living 20 days per scene. We aggregate daily intentions and tasks into one data point per human. Two 10-way BERT-large-uncased classifiers [11] are finetuned—one for intentions (10 epochs), one for tasks (20 epochs) with train-test split 0.8:0.2, learning rate 5e-6, and tested on an unseen scene. As shown in Table. 1, task classification is harder than intention classification, as intentions align more with human traits, but tasks (e.g., drinking water) may correspond to multiple intentions (e.g., leisure, exercise).

**Diverse Simulated Humans.** We assess diversity by standard deviation (SD) of Big-5 traits (1–5 scale) across 10 simulated humans. We compute per-trait SD and take average. From Table. 1, the high SD exceeds the typical 0.7–0.9 range in real-world distributions [62], validating diversity.

**Human Traits and Psychometrics Coherence.** In our main approach, the robot-VLM infers Big-5 traits throughout the day based on the human's intention and task history. Using the final scores at the end of collaboration, we assess coherence with ground truth via Pearson correlation [48, 4], and introduce a one-step mismatch for comparison. The significant drop in correlation for mismatched pairs (Table. 1) confirms alignment between inferred traits and psychometric data, demonstrating the LLM's ability to interpret human psychology from behavior.

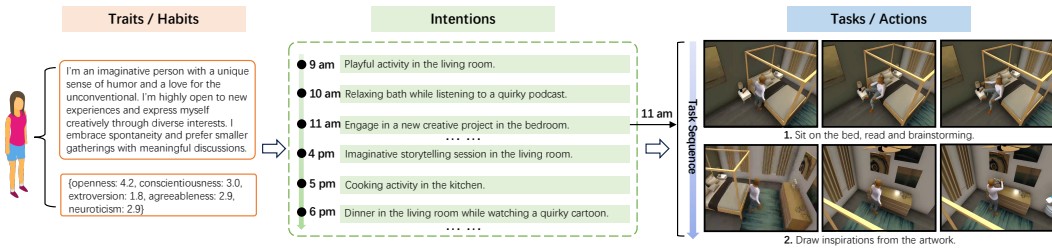

Figure 5: **Qualitative examples of full-day intentions and tasks** proposed by a human with specific human traits and psychometric data.

**Temporal Dependence in Human Behavior.** We study how current-hour intentions depend on prior hours via a next-sentence prediction task: given three earlier intentions (e.g., 9–11 am), the 12 pm intention is used as the positive example, with other-time intentions as negatives. We evaluate on 10 simulated humans, each living for 20 days in 5 scenes. BERT-large-uncased [11] is trained for 20 epochs (learning rate 5e-6). Results in Table. 1 confirm temporal dependence.

**User Studies.** We conduct two user studies (25 participants each) to assess: 1) Whether real humans can identify the same simulated human across days and scenes, and 2) Whether real humans can distinguish simulated humans with varying traits and Big-5 scores. For 1), we sample full-day intentions and tasks of 10 simulated humans across 2 days and 2 scenes (4 samples/human), then construct 10 multiple-choice questions showing a human profile and three behavior options (1 correct, 2 distractors). For 2), we sample full-day behaviors from 10 simulated humans with distinct traits, and ask participants to match trait descriptions to the corresponding full-day intentions and actions. Results in Table. 1 show higher accuracy in identifying the same simulated human than in distinguishing between different simulated humans. This discrepancy likely arises because multiple-choice tasks provide explicit answer options, reducing ambiguity, whereas trait-based matching requires deeper reasoning about personality-behavior relationships, making it more challenging.

**Alignment with Real-Human Behavior.** We study how simulated human intentions align with real-human intentions. We recruit six participants who provide personality traits and psychometric data, and record their daily intentions over five days. Using these traits, we prompt the LLM to generate simulated intentions for the same time span. To

Table 2: **Semantic alignment** between simulated and real-human intentions.

|  | Generic | Mismatched | Main |
|---|---|---|---|
| SBERT ↑ | 0.554 | 0.523 | 0.810 |
| OpenAI Emb. ↑ | 0.537 | 0.543 | 0.772 |

assess alignment, we aggregate both sets into single paragraphs (removing time formatting like "9am: ..." to avoid inflated structural similarity) and compute semantic similarity using SBERT (all-mpnet-base-v2) [57] and OpenAI embeddings (text-embedding-3-small) [1]. We compare against: 1) prompting without human profile (generic) and 2) mismatched LLM-human intention pairs (mismatched). From Table 2, both baselines yield moderate similarity (∼0.5), as sentence encoders assign partial similarity to structurally similar content. In contrast, aligned pairs achieve much higher scores, indicating strong alignment between simulated and real human intentions. SBERT slightly outperforms OpenAI embeddings, likely due to its sentence-level training objective.

**Qualitative Results.** We present examples of full-day intentions and tasks proposed by a human with specific human traits and psychometric data in Fig. 5.

### 4.3 Analysis of Continual, Open-Ended HRC

Since COOPERA involves long-term, open-ended tasks that requires the robot reasoning over human traits and temporal context, we construct baselines using standard LLM/VLM-based approaches adapted for task inference. These baselines reflect commonly used paradigms in open-world robot planning [8, 29].

**Baselines. 1) Direct Prompting:** The robot proposes a single intention from visual input and decomposes it into tasks. The robot-VLM is optimized solely via prompting with retrieved intention/task history. **2) Direct Finetuning:** The robot brain is finetuned to directly output a single intention and decompose it into tasks. **3) Oracle:** The robot is given the ground-truth human intention and decomposes it into tasks. **4) Random:** Intention and task classifiers are removed; all proposed intentions and tasks are accepted without validation. **5) Intention Agnostic:** The robot directly

predicts and filters tasks without first inferring intentions. **6) Human** & **Context Agnostic:** The classifiers do not learn the correlation between human traits and intentions/tasks or the temporal dependence between previous and current intentions/tasks. They only learn the relationship between the current time and the intentions/tasks.

**Evaluation Metrics.** We assess the assistive agent's performance using F1-based success rate across three methods, ensuring a comprehensive evaluation from simulation to real-world perspectives, incorporating prior HRC approaches [51, 8]. **1) Predicate-based:** Tasks are executed and evaluated by predicate functions with success based on object class matches rather than instance matches, following Watch-and-Help [50]. **2) LLM-based:** Given the ground truth human intention and predicted tasks, the human-LLM judges whether each task fulfills the intention (binary yes/no), and F1 is computed over these labels. **3) Human verification:** Same as 2) but evaluated by real human users. In our main method, the robot-VLM generates a task superset, which is filtered by a task classifier assigning yes/no labels used for F1 computation. For baselines without a task classifier (e.g., Direct Prompting), all predicted tasks are treated as positive. Please see Appendices C and D for details on predicate construction and LLM evaluation.

**Setup.** We follow the evaluation settings in Section 3. Setting 1 evaluates 5 humans across 2 scenes; Setting 2 evaluates 10 humans; Setting 3 spans 3 distinct scenes; and Setting 4 spans 9 humans.

**Analysis of Assistive Performance.** We analyze two aspects: 1) Within-day improvement—does the robot's collaboration success increase throughout the day by learning temporal dependencies between human intentions and actions? 2) Across-day improvement—does collaboration become more successful and personalized over multiple days, using end-of-day feedback? From Fig. 6 (a), our method achieves the highest within-day improvement. In contrast, prompting, random, and oracle exhibit little to no improvement, or even decline. We hypothesize that these methods do not

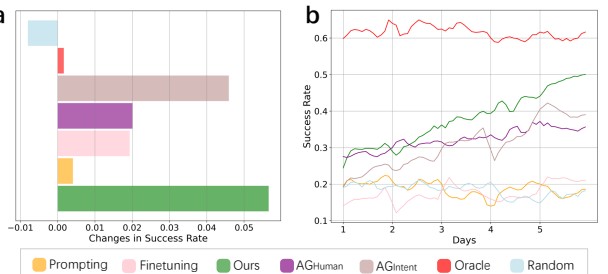

Figure 6: **Evaluation of changes in robot success rate (predicate-based):** (a) within a single day and (b) across multiple days. See Appendix E for a detailed breakdown of the results and additional analysis.

benefit from learning human intentions, which are highly correlated with human traits and temporal context. This finding aligns with our human classification experiment in Section 4.2. From Fig. 6 (b), our method shows the strongest improvement across days, second to oracle. The minimal gain in prompting and finetuning highlights the challenge of varying human behavior across days, as these methods tend to establish a 1-to-1 mapping between time and human intentions/tasks. Prompting relies heavily on collaboration history, while finetuning prioritizes the highest probability training data, limiting adaptability to varying human behaviors.

**Out-of-Domain Generalization.** We study **1) Scene generalization:** can the robot personalize collaboration with a human in an unseen scene after interacting in other scenes? **2) Human generalization:** can the robot collaborate effectively with a new human after training with others? For 1), we use models finetuned from setting 2 on four scenes and evaluate on a fifth, unseen scene with the same human. The baseline is the robot's average performance during its initial interaction in the unseen scene, without finetuning on the previous scenes, averaged over 10 humans.

Table 3: **Generalization performance.** We report the average success rate (predicate-based).

|        | Baseline | Finetuned |
|--------|----------|-----------|
| Scene  | 0.269    | 0.465     |
| Human  | 0.258    | 0.343     |

For 2), we use models finetuned from setting 3 on three humans and evaluate on collaboration with a fourth unseen human. The baseline is the robot's unadapted performance when first interacting with the new human, without finetuning on the previous three, averaged over 5 new humans. Result in Table. 3 show that generalizing to a new human is much harder than to a new scene. This is likely due to greater variability in human behaviors. While the robot can learn shared patterns across humans, effective collaboration with a new human requires adaptation to fine-grained, person-specific traits.

**Qualitative Results.** We show how the robot improves collaboration within a day by inferring more correct tasks for assistance, along with a visualization of HRC at a specific time in Fig. 7.

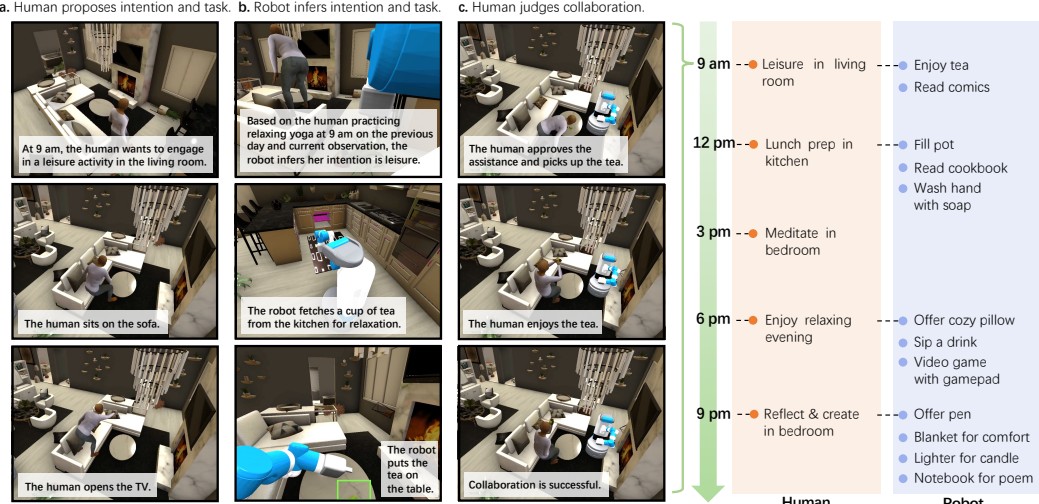

**a.** Human proposes intention and task.  **b.** Robot infers intention and task.  **c.** Human judges collaboration.

Figure 7: **Qualitative examples of successful human-robot collaboration within one day.** The red column displays human intentions, while the blue column shows the robot's correctly inferred tasks for assistance.

## 4.4  Analysis of Real-World Applicability

The ultimate goal of COOPERA is to develop robot agents that assist real humans by adapting to their preferences over long-term. Yet, fully real-world experiments pose ethical, safety, and cost issues (requiring a real human and physical robot to interact in a household over multiple days). To validate applicability of COOPERA under real-world conditions, we take three complementary approaches. Please see Appendix E for detailed results, qualitative examples, and additional analysis.

**Human Verification.**  We validate if predicate-based and LLM-based evaluations align with human verifications by sampling one episode per setting from our main method. From Table. 4, the low L1 indicates strong correlation among all three metrics, supporting our framework's real-world applicability. This justifies our use of VLMs/LLMs for predicting human actions in both baselines and main method, as their reasoning closely resembles human decision-making.

Table 4: **Correlation between predicates, LLM, and human evaluations (rows 1–2):** L1 ↓, averaged over collaboration types. **Offline real-human and human-in-the-loop collaboration (row 3–4):** predicate-based success rate (SR ↑), averaged over the final day.

|  | Setting 1 | Setting 2 | Setting 3 | Setting 4 |
|---|---|---|---|---|
| Predicate vs. Real-Human (L1 ↓) | 0.091 | 0.091 | 0.085 | 0.120 |
| LLM vs. Real-Human (L1 ↓) | 0.077 | 0.080 | 0.077 | 0.075 |
| Offline Real Human (SR ↑) | 0.498 | 0.471 | 0.426 | 0.322 |
| Human-in-the-Loop (SR ↑) | 0.488 | 0.467 | 0.431 | 0.349 |

Also, in open-ended settings with large state spaces, LLMs/VLMs serve as effective reasoning modules due to their generalization abilities [8, 29].

**Collaborating with Offline Real Humans.**  Real humans exhibit greater behavioral dynamics and emergent decisions due to temporary factors (e.g., plans, mood, weather). We recruit six participants who provide personality traits and psychometric data, and record their daily intentions over five days. An LLM decomposes these into tasks in HSSD scenes, where the robot collaborates across all settings. Results in Table 4 show performance comparable to simulated humans (Table. 6 row 4), despite increased dynamics.

**Human-in-the-Loop.**  Six real humans replace the LLM and collaborate with the assistive agent. They are shown retrieved object and motion sets and select which to interact with based on their intention. For ease of implementation and usability, participants input responses as text rather than using a keyboard to control the simulated agent. Results in Table 4 indicate that real human collaborators do not make the task more difficult and can, in some cases, lead to higher success rates compared to offline or simulated humans.

## 4.5  Ablation Studies

**Human Simulation.  1) Removing human profile extension:** To explore whether our method yields the most distinct simulated humans, we remove simulated conversations and profile extension, prompting LLM only with the original short trait paragraph. We evaluate by finetuning and measuring

Table 5: **Ablation study on human simulation.** The effects of removing profile extension on human classification and using single-shot intention proposal on temporal dependence.

| | Removing Profile Extension (Acc ↑) | | All-Day Intention Proposal | |
| | intention cls. | task cls. | Acc ↑ | F1 ↑ |
|---|---|---|---|---|
| Removed | 0.950 | 0.800 | 0.751 | 0.740 |
| Ours | 0.995 | 0.830 | 0.789 | 0.790 |

Table 6: **Ablation study on assistive agent:** success rate (predicate-based) ↑ averaged over the last day.

| | Setting 1 | Setting 2 | Setting 3 | Setting 4 |
|---|---|---|---|---|
| No Traits | 0.481 | 0.443 | 0.239 | 0.206 |
| No Context | 0.452 | 0.414 | 0.408 | 0.299 |
| Changing Backbone | 0.487 | 0.424 | 0.362 | 0.310 |
| Ours (main) | 0.505 | 0.465 | 0.439 | 0.344 |

classifiers accuracy (Section 4.2). **2) Single-shot human intention proposal:** We test if our pipeline design maximizes temporal dependence in human behavior. Instead of proposing intentions hour by hour with access to intention/task history and using Reflexion, we remove these and generate all intentions at once. We evaluate via next-intention prediction (Section 4.2). Results in Table. 5 confirm the effectiveness of the profile extension module and the overall pipeline design.

**Assistive Agent.** **1) Removing human traits inference.** We examine the importance of learning human traits by removing robot's inference of human traits or Big-5 scores from past intentions and tasks, preventing classifiers from learning their correlation. **2) Removing temporal context learning:** We assess the impact of learning temporal dependence between human intentions and tasks by preventing classifiers from using past intentions/tasks when predicting the current one. **3) Changing the robot brain backbone:** We replace the robot-VLM from Llama-3.2-11B [13] to LLaVA-1.6-Mistral-7B [34]. Using different VLMs for human and robot reduces alignment and tests robustness. From Table 6, removing trait inference significantly reduces success rate in settings 3 and 4 involving multiple humans, as the robot struggles to distinguish them. The learning of time-based context benefits all settings. Despite smaller model size, the robot still achieves reasonable success.

# 5 Conclusion

We introduce **COOPERA**, a framework for continual, open-ended HRC. We propose a human model to generate long-term human behaviors driven by personality traits, a benchmark, and a method to assist humans under by predicting their long-term intentions. Our framework opens up exciting directions for future work, such as using communication to better infer human traits or build agents that can perform proactive assistance (e.g. arranging a house before the start of the day based on preferences). We hope that this work can promote future research on building agents that can work over long time horizons and adapt to human preferences.

**Limitations.** Despite compelling HRC performance, COOPERA currently focuses on single-human settings, leaving multi-human collaboration for future work. While evaluations are primarily conducted in simulation, we validate sim-to-real transfer through real-human routines and interactions. Our method uses skill primitives compatible with standard robot platforms (e.g., Fetch), making it readily transferable to real hardware.

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

# Appendix for *COOPERA*

## A Overview

This Appendix includes: 1) more details about how we construct dynamic HSSD scenes and statistics, 2) additional details of how we build, train, and evaluate the simulated human and the assistive agent, 3) additional HRC results and analysis, including both quantitative metrics and qualitative examples, along with comparisons to human verification of our main method and baselines across collaboration types and settings, and 4) prompt details.

## B Additional Details of Scenes

### B.1 Dynamic Habitat Synthetic Scenes Dataset Construction

We make certain objects in the HSSD [28, 40, 53, 54] scenes dynamic by checking if they are supported by any structure and their object super categories. For objects that do not have support, we classify them as follows:

**dynamic_categories** = [trashcan, decor, dining ware, plant, electronics, animate object, apparel, liquid container, kitchen ware, tray, bathroom accessory, gym equipment, toy, wearable]

**static_categories** = [storage furniture, support furniture, seating furniture, floor covering, lighting, sleeping furniture, bathroom fixtures, mirror, large kitchen appliance, large appliance, kitchen bathroom fixture, vehicle, heating cooling, medium kitchen appliance, display, arch, curtain, small kitchen appliance]

### B.2 Scene Statistics

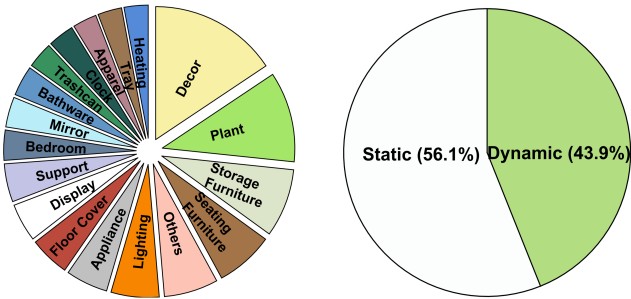

Figure 8: **Distribution of object categories** within the constructed dynamic HSSD scenes.

Our selected 5 scenes feature varying number of rooms ($4-11$), static objects ($51-140$), and dynamic objects ($33-94$). We present the distribution of 18 representative object categories (out of 32) in Fig. 8. The large number and diversity of objects and categories enable humans to propose a wide range of open-ended tasks.

### B.3 Scene Summarization and Visualizations

We summarize 3D environment information in each scene as a text-based dictionary, which is used as input to the simulated humans. Specifically, we extract the bounding boxes of rooms and map each object to its corresponding room, forming an object-room mapping in the format of object_ID: [object_name, room]. For example:

**mapping** = {'Nemo Kepler Pendant, Black': [125, 'corridor'], 'AquaVive stortdoucheset Kila met kraan': [123, 'main bathroom'], 'Uttermost Marlow Chandelier': [118, 'bathroom of bedroom 1'], 'Eisa Pendant': [114, 'main bedroom'], 'CAR - SUV': [12, 'garage'], 'Nolan Upholstered King Bed': [109, 'main bedroom'],  . . . }

We show visualizations of the five scenes used in COOPERA in Fig. 9.

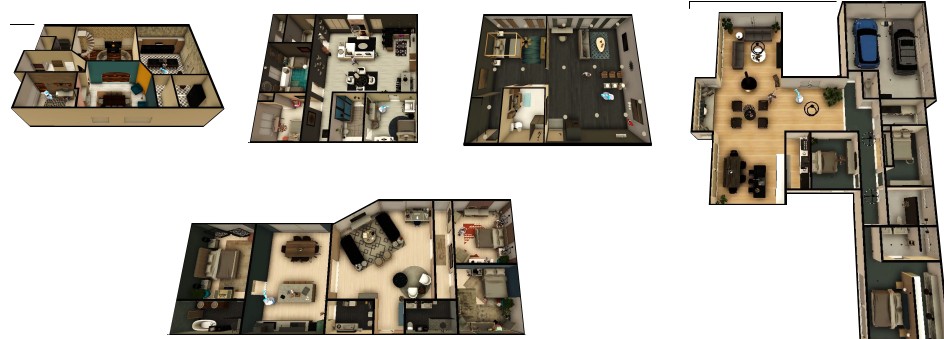

Figure 9: **HSSD Scenes used in COOPERA.**

# C Additional Details of Human Simulation

## C.1 Psychometric Data Computation

Since SPC dataset lacks psychometric data, we derive Big-5 OCEAN scores by prompting the LLM to 1) directly infer the scores [48] and 2) complete the Big-5 personality test [18, 45] and compute scores based on the formula. We then take a majority vote across five inference trials, using bins of 0.5 on a scale of 1-5.

## C.2 3D Motion

For human simulation, free-form motion data is formatted in SMPL-X, enabling detailed control over whole-body motions, including facial expressions and finger articulation. To integrate this data into the human simulation pipeline of Habitat 3.0 [52] , we remap the format of Motion-X [31] data, as illustrated below.

**global root orientation**: SMPL-X[:, :3]
**body**: SMPL-X[:, 3:3+63]
**finger articulation**: SMPL-X[:, 66:66+90]
**yaw pose**: SMPL-X[:, 66+90:66+93]
**face expression**: SMPL-X[:, 159:159+50]
**global body position**: SMPL-X[:, 209:209+100]
**global body position**: SMPL-X[:, 309:309+3]
**body shape**: SMPL-X[:, 312:]

## C.3 Feedback

Please refer to Section F for details on our designed feedback mechanism for the robot's predicted assistive tasks. We emphasize that while we structure the feedback as binary answers paired with reasoning, it can be easily adapted for other learning algorithm by modifying the prompt.

## C.4 Additional Qualitative Examples

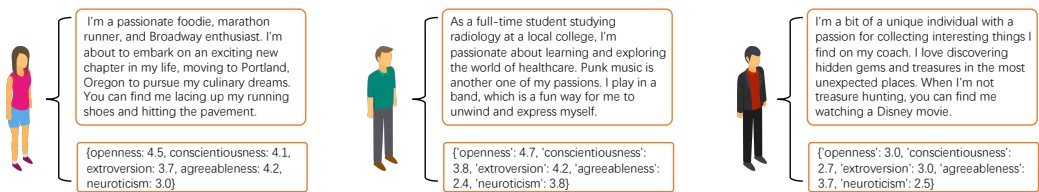

Figure 10: **Additional examples of simulated human profiles psychometric data.**

We present additional examples of simulated human profiles along with their corresponding psychometric data in Fig. 10.

# D Additional Details of Building and Evaluating an Assistive Agent

## D.1 Navigation and Movement

To navigate the scene, the agent uses classical motion planning for path calculation, with a stopping threshold determined by the object's Axis-Aligned Bounding Box (AABB). For the Fetch robot's arm movements, inverse kinematics (IK) is applied to calculate the end effector (EE) position based on the 3D location of the target object. Robot actions are implemented using primitives from Habitat's `task_action` registry, combining navigation and EE control to form a full pick-and-place pipeline: 1) `MoveEEAction` moves the EE via Cartesian steps with IK; 2) `PickObjIdAction` grasps objects using a snap mechanism within a distance threshold; 3) `PlaceObjIdAction` releases objects via desnap; and 4) `ResetEEAction` resets the EE to a default pose.

## D.2 Training

Task videos from Habitat 3.0 [52] are resized to $1024 \times 768$ and input to Llama-3.2-11B [13] (our robot-VLM) using Llama's default settings. Classifiers are finetuned on Mistral-7B-Instruct-v0.2 [27] using LoRA [22] (rank 8, dropout 0.2, alpha 16; targets: q, k, v, o) in an instructional format to output binary yes/no. We train for 5 epochs using AdamW [37] (lr 1e-5, weight decay 0.01), with batch size 1 and gradient accumulation of 4 steps, across 3 NVIDIA A10 GPUs (24GB RAM).

For training the robot's intention and task binary classifiers via instructional finetuning of an LLM, we structure the input data in the following format:

```
### Instruction:
Considering the human's profile, traits, temporal dependence on past behaviors, and the current time,
determine if it is likely or unlikely that this human will:  ...  Respond with 'Yes' or 'No'.

### Input:
Human Profile.
Big Five Traits.
Previous Relevant Intentions.
Previous Relevant Tasks.
Current Time.

### Response:
```

## D.3 Predicate Construction

| Collaboration Type 1 | Time: 6 pm
Intention: Enjoy a playful and imaginative dinner preparation in the kitchen, experimenting with new recipes and flavors. | Task 1: Place the "tuna_fish_can" on the "KITCHEN AID ARTISAN MIXER RED" for setup.
Task 2: Place the "mustard_bottle" on the "Edelweiss desk, ash/white" in the living room.
Task 3: Place the "fork" on the "Edelweiss desk, ash/white" in the living room. | ON(food, kitchen electricals)

ON(food, table)

ON(kitchen items, table) |
|---|---|---|---|
| Collaboration Type 2 | Time: 9 am
Intention: Engage in a morning workout routine in the living room. | Task 1: Start with a warm-up by performing yoga stretches near the "Marina Slipcover Sofa" to prepare the body for more intense exercises.
Task 2: Perform squats with dumbbells near the "Wendover Art # Morning Lake View I" for strength training.
Task 3: Transition to cardio by doing jumping jacks near the "Wendover Art # Off the Path" to elevate heart rate.
Task 4: Engage in core exercises by doing planks near the "APPLE iMac 5K 27" to strengthen abdominal muscles.
Task 5: Cool down with a stretching routine near the "Marina Slipcover Sofa" to relax muscles and prevent injury. | NEED(yoga mat)

NEED(dumbbells)

NEED(jump rope)

NEED(exercise mat)

NEED(towel) |

Figure 11: **Examples of predicate construction.**

For collaboration type 1, we map objects to Habitat 3.0 and the YCB dataset's predefined semantic categories, allowing evaluation via exact match. For collaboration type 2, where the robot offers objects from a magic box, we compute semantic similarity between the robot's predicted tasks/objects for assistance and the human's desired objects, using a threshold of 0.6. Fig. 11 illustrates examples of predicate construction.

# E  Additional Experiments and Details

## E.1  Results Breakdown and Analysis

In Fig. 12, we show detailed predicate-based and LLM-based evaluations in both collaboration types across all collaboration settings. Overall, LLM-based evaluations yield higher scores than predicate-based ones, particularly in settings 3 & 4, which are more challenging. This suggests that predicate-based evaluations are more rigorous, requiring exact matches for collaboration type 1 and high semantic similarity for collaboration type 2. We highlight several key findings and analyze the performance of each baselines in detail below.

**Temporal Fluctuation Patterns Across Days.**  In Fig. 12, we observe a consistent temporal pattern across days: performance tends to dip around midday and recover toward the end of the day, and improve further on the following day. We attribute this to two primary factors. First, it reflects how human routines are structured, both in reality and in our simulation. Humans generally engage in more predictable activities in the morning and evening (e.g., hygiene, eating, relaxing), which are easier for the robot to infer and assist with. In contrast, midday behavior tends to be more diverse and more strongly influenced by individual traits. Since our human simulation pipeline samples intentions and tasks based on both traits and time, this increases the diversity of midday actions, making inference and alignment more difficult for the robot and resulting in a temporary performance drop. Performance recovers as routine behaviors re-emerge in the evening or the following morning. Importantly, despite these fluctuations, overall performance improves across days. Second, the extent of this fluctuation increases with task difficulty, as defined in our framework (Section 3.1). In Setting 1 (same human, same scene), the robot benefits from consistent exposure to both the user and environment, resulting in relatively small midday dips. In Setting 2 (same human, different scenes), unfamiliar object layouts introduce grounding challenges that increase midday variability. In Setting 3 (different humans, same scene), rotating between users interrupts personalization, making intention interpretation more difficult—particularly around midday when behavior is less routine. Finally, Setting 4 (different humans, different scenes) combines both challenges, resulting in the largest fluctuations as the robot must generalize across both human and spatial contexts.

**Performance Analysis: Main Method vs. Baselines.**  We evaluate our method against six baselines: 1) Direct Prompting, 2) Direct Fine-tuning, 3) Oracle, 4) Random, 5) Intention Agnostic, and 6) Human & Context Agnostic. Fig. 12 shows that our method achieves both the highest adaptation trend over time and the highest final success rate (second only to Oracle) by explicitly modeling the correlation between human intentions/tasks, traits, and temporal dependencies. As the day progresses, the robot accumulates interaction history and adapts its behavior to the human preferences. 1) Direct Prompting shows minimal improvement as it depends on retrieved interaction history without learning the human preferences. 2) Direct Finetuning performs slightly better but still struggles due to its rigid mapping from inputs to intentions/tasks, making it biased toward frequent training examples and overlooking the varying distribution of human behavior. 3) Oracle serves as an upper-bound performance reference by receiving ground-truth human intentions. While this provides a significant advantage, its performance is static by design. This is because Oracle bypasses the core challenge of accurately mapping visual observations of humans performing detailed tasks to high-level intentions, a fundamental problem in human-robot collaboration, even in closed-set scenarios. Moreover, since the correct intention is provided upfront, Oracle has no need to learn temporal dependencies between human intentions (e.g., an intense workout at 11 am typically leads to lunch preparation at 12 pm). 4) Random degrades over time due to the absence of learning or validation. 5) Intention Agnostic shows moderate improvement but lags behind our method, as it skips intention inference and thus loses contextual information, which is problematic since temporal dependencies between intentions are typically stronger than between tasks (validated in Section 4.2). 6) Human & Context Agnostic captures only superficial time-task correlations and ignores human traits or temporal context, leading to weak within-day gains.

## E.2  Human Verification Breakdown and Analysis

For human verification, we present additional results across three baselines and our main method, analyzing how human evaluations align with predicate-based (Table. 7) and LLM-based (Table. 8) assessments in a detailed breakdown. Due to cost constraints, we use a subset of data (one episode per setting), resulting in 32 episodes across the four methods. We recruit eight human evaluators, each

Table 7: **Breakdown of correlation between predicates and human evaluations.** We report the L1 ↓.

| | Collaboration 1 | | | | Collaboration 2 | | | |
| --- | --- | --- | --- | --- | --- | --- | --- | --- |
| | **Setting 1** | **Setting 2** | **Setting 3** | **Setting 4** | **Setting 1** | **Setting 2** | **Setting 3** | **Setting 4** |
| Random | 0.070 | 0.134 | 0.067 | 0.065 | 0.073 | 0.079 | 0.088 | 0.112 |
| Oracle | 0.072 | 0.098 | 0.084 | 0.065 | 0.086 | 0.102 | 0.104 | 0.108 |
| Human & Context Agnostic | 0.062 | 0.121 | 0.066 | 0.071 | 0.073 | 0.109 | 0.080 | 0.100 |
| Main | 0.092 | 0.096 | 0.094 | 0.087 | 0.090 | 0.085 | 0.076 | 0.153 |
| Average | 0.074 | 0.112 | 0.078 | 0.072 | 0.080 | 0.094 | 0.087 | 0.118 |

Table 8: **Breakdown of correlation between LLM and human evaluations.** We report the L1 ↓.

| | Collaboration 1 | | | | Collaboration 2 | | | |
| --- | --- | --- | --- | --- | --- | --- | --- | --- |
| | **Setting 1** | **Setting 2** | **Setting 3** | **Setting 4** | **Setting 1** | **Setting 2** | **Setting 3** | **Setting 4** |
| Random | 0.031 | 0.054 | 0.066 | 0.044 | 0.068 | 0.091 | 0.084 | 0.110 |
| Oracle | 0.103 | 0.096 | 0.087 | 0.043 | 0.071 | 0.069 | 0.091 | 0.094 |
| Human & Context Agnostic | 0.045 | 0.058 | 0.076 | 0.071 | 0.084 | 0.089 | 0.079 | 0.108 |
| Main | 0.063 | 0.075 | 0.081 | 0.073 | 0.090 | 0.085 | 0.072 | 0.076 |
| Average | 0.061 | 0.071 | 0.078 | 0.058 | 0.078 | 0.084 | 0.082 | 0.097 |

assessing four randomly assigned episodes. The results show that the gap between predicate-based evaluation and human verification is larger than that between LLM-based evaluation and human verification. This is because LLMs/VLMs reason and make judgments for assistance in a way that aligns more closely with human preferences and decision-making, whereas predicate-based evaluation relies on strict matching criteria that may not fully capture nuanced human reasoning.

### E.3 Qualitative Examples of Offline Real Human

We present qualitative examples of offline real humans in Fig. 13. Compared to simulated humans (Fig. 5), real humans exhibit greater behavioral dynamics and emergent decisions due to temporary factors (e.g., plans, mood, weather).

### E.4 Analysis of Human-in-the-Loop Collaboration

Intuitively, one might expect lower performance when collaborating with real humans due to their inherent variability and spontaneity. However, results in Table 4 indicate that real human collaborators do not make the task more difficult and can, in some cases, lead to higher success rates compared to offline or simulated humans. Through discussions with participants, we identify a likely explanation. Unlike the offline setting where participants record their routines throughout an entire day (12 one-hour intervals covering the time from 9 am to 9 pm), the HITL experiments take place in a short, controlled session. As a result, participants do not act spontaneously hour-by-hour but instead recall and follow their typical weekly routines from memory, which tend to be more consistent. This reduced behavioral variability allows the robot to personalize more quickly. and further validates the behavioral diversity modeled in our simulated humans.

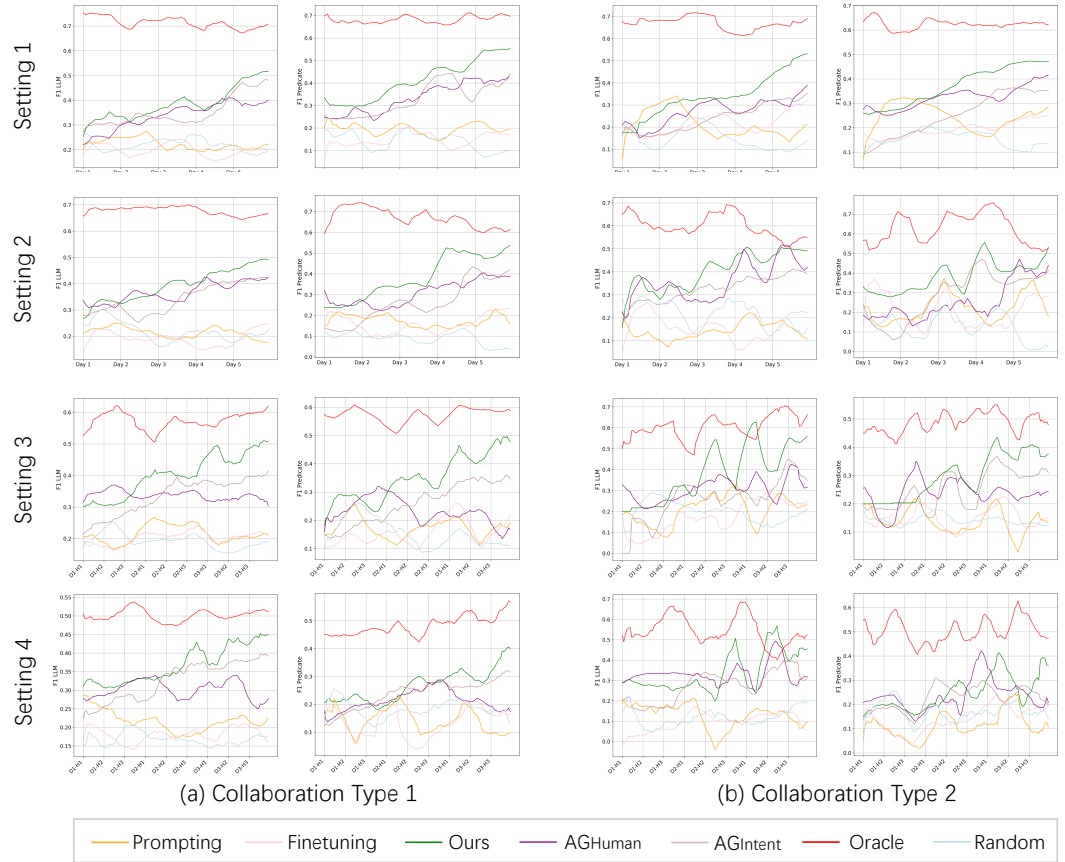

Figure 12: **Breakdown results of HRC.** We compare our main method with the baselines across days.

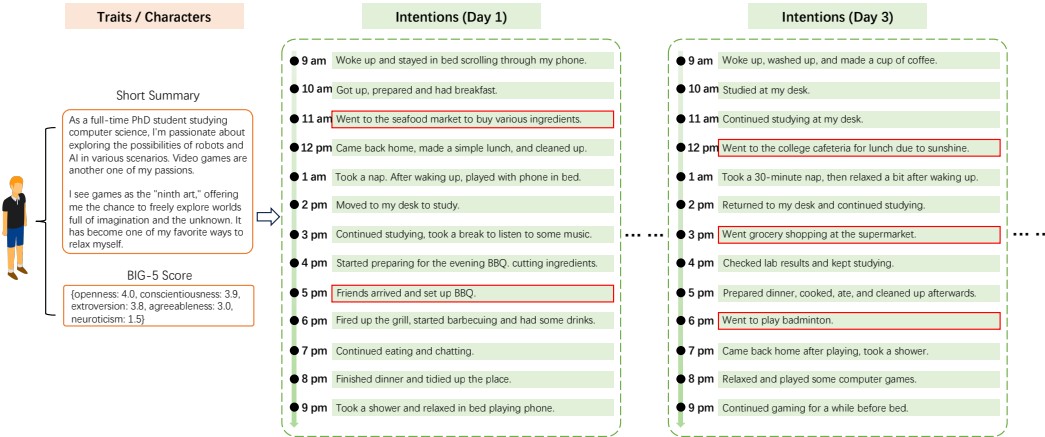

Figure 13: **Qualitative examples of full-day intentions** recorded by a real human with specific human traits and psychometric data. Red boxes highlight emergent behaviors due to temporary factors (e.g., plans, mood, weather).

# F    Prompt Details of the Simulated Human

We show exact prompts for LLMs in simulated human behavior.

---

**Human Profile Summary and Extension.**

Input:
1. Human 1 profile.
2. Human 2 profile.
3. Conversation between Human 1 and Human 2.

Instruction:
Summarize and reasonably expand Human 1's profile into a first-person self-introduction based on their initial profile and past conversations with other humans. Provide a detailed description covering, if presented, the person's job, hobbies, daily activities, food preferences, social life, physical activity, entertainment preferences, travel habits, personal values, goals and aspirations, stress and coping mechanisms, technological use, cultural interests, health and wellness, community involvement, education, financial habits, and personal style.

---

**Human Psychometric Data Computation (LLM Inference).**

Input:
1. Human profile.

Instruction:
Infer Big Five personality traits (scale 1-5, float) based on the provided human profile. Write in the following format: {'openness': a, 'conscientiousness': b, 'extroversion': c, 'agreeableness': d, 'neuroticism': e}

---

**Human Psychometric Data Computation (Big-5 Personality Test).**

Input:
1. Human profile.

Instruction:
Based on the provided human profile, complete the Big-5 personality test. In the questions below, for each statement 1-50 mark how much you agree with on the scale 1-5, where 1=disagree, 2=slightly disagree, 3=neutral, 4=slightly agree and 5=agree.

Questions:
1. Am the life of the party.
2. Feel little concern for others.
. . . . . .
50. Am full of ideas.

---

**Human Intention Proposal.**

Input:
1. Current time.
2. A list of rooms in the house (ignore small spaces like closets).
3. Your Big Five scores (scale 1-5) and human profile.
4. Most relevant human intentions proposed at previous times (ignore if empty—this means it's the first intention of the day).
5. Most relevant human tasks proposed at previous times.ids (ignore if empty—this means it's the first task of the day).

You are a human living in the house. Propose your intention at the current time.

Instructions:
1. Intention must align with your Big 5 scores, reflect all aspects of the profile, and be diverse yet reasonable based on the house layout and available objects.

2. Intention must be high-level and either human-centric (e.g., hygiene, sport, leisure) or room-centric (e.g., clean, organize, set-up). Do not mention specific objects.
3. Intention must have temporal dependence but be non-repetitive with the previous intentions and tasks.
4. Intention should be within the house.
5. All objects are rigid and cannot deform, disassemble, or transform.

Write in the following format. Do not output anything else:
Time: xxx am/pm (e.g., 9 am)
Intention: basic descriptions.
Reason_human: detailed descriptions of why it follows your Big 5 scores and profile.
Reason_intentions: detailed descriptions of why it has temporal dependence with the previous, relevant intentions at [list of time].
Reason_tasks: detailed descriptions of why it has temporal dependence with the previous, relevant tasks at [list of time.id].

## Human Task Proposal.

Input:
1. The proposed intention at current time.
2. A dict mapping rigid, static objects to their IDs and rooms.
3. Your Big Five scores (scale 1-5).
4. Most relevant human intentions proposed at previous times (ignore if empty—this means it's the first intention of the day).
5. Most relevant human tasks proposed at previous times.ids (ignore if empty—this means it's the first task of the day).

You are a human living in the house.

Instructions:
1. Break down the intention into 5 tasks for collaboration with a robot.
2. Task types:
- Type 1: Creative, reasonable free-form human motion interacting or approaching a fixed, static object (static objects cannot be moved) with an object in hand provided by the robot (e.g., sit on sofa with TV remote control in hand, wipe table with tissue in hand, squat with dumbbell in hand near rug).
3. For interacting with fixed, static objects, use only objects from the given static object dict. For objects in hand, a robot will provide them.
4. Both interacting and inhand objects must be specified (cannot be none).
5. Tasks should be continuous and logical, and align with your Big 5 scores and profile.
6. Tasks must have temporal dependence with the intentions and tasks at previous times.
7. Free-form motion should be diverse. Examples: sampled_motion_list. Feel free to propose others.
8. All objects are rigid and cannot deform, disassemble, or transform.

Write in the following format. Do not output anything else:
Time: xxx am/pm
Intention: basic descriptions.
Tasks:
1. Thought: detailed descriptions of the task. Reason_human: why it aligns with your Big 5 scores and profile. Reason_intentions: how it depends on previous, relevant intentions at [list of time]. Reason_tasks: how it depends on previous, relevant tasks at [list of time.id]. Act: [type: 1, inter_obj_id: real int, inter_obj_name: xxx, inhand_obj_name: yyy, motion: free-form motion]
2. ...

## Human Reflection (Profile).

Input:
1. The proposed intention at current time.
2. A dict mapping rigid, static objects to their IDs and rooms.

3. Your Big Five scores and human profile.
4. Most relevant human intentions proposed at previous times (if empty, ignore it—this means it's the first intention of the day).
5. Most relevant human tasks proposed at previous times.ids (if empty, ignore it—this means it's the first intention of the day).

Your task is to check if the temporal dependence and human profile are strictly followed in each task, and revise to make better if necessary.

Instructions:
1. Tasks should be continuous and logical, and align with your Big 5 scores and profile.
2. Tasks must have temporal dependence with the previous intentions and tasks, with detailed explanation mentioning previous intentions and tasks explicitly.
3. For interacting with fixed, static objects, use only objects from the given static object dict. For objects in hand, a robot will provide them.

Write in the following format. Do not output anything else:
Time: xxx am/pm
Intention: basic descriptions.
Reflect Each Task:
1. no mistake or change made.
2. ...
Revised Tasks:
1. Thought: detailed descriptions of the task. Reason_human: why it aligns with your Big 5 scores and profile. Reason_intentions: how it depends on previous, relevant intentions at [list of time]. Reason_tasks: how it depends on previous, relevant tasks at [list of time.id]. Act: [type: 1, inter_obj_id: real int, inter_obj_name: xxx, inhand_obj_name: yyy, motion: free-form motion]
2. ...

## Human Reflection (3D Info).

Input:
1. The proposed intention at current time.
2. A dict mapping rigid, static objects to their IDs and rooms.

Your task is to check if the instructions are strictly followed in each task, and revise to make better if necessary.

Instructions:
1. Break down the intention into 5 tasks for collaboration with a robot.
2. Task types:
- Type 1: Creative, reasonable free-form human motion interacting or approaching a fixed, static object (static objects cannot be moved) with an object in hand provided by the robot (e.g., sit on sofa with TV remote control in hand, wipe table with tissue in hand, squat with dumbbell in hand near rug).
3. For interacting with fixed, static objects, use only objects from the given static object dict (exact name). For objects in hand, a robot will provide them.
4 Both interacting and inhand objects must be specified. Importantly, they cannot be none.
5. Free-form motion should be diverse. Examples: sampled_motion_list. Feel free to propose others.
6. All objects are rigid and cannot deform, disassemble, or transform.

Write in the following format. Do not output anything else:
Time: xxx am/pm
Intention: basic descriptions.
Reflect Each Task:
1. no mistake or change made.
2. ...
Revised Tasks:

1. Thought: detailed descriptions of the task. Reason_human: why it aligns with your Big 5 scores and profile. Reason_intentions: how it depends on previous, relevant intentions at [list of time]. Reason_tasks: how it depends on previous, relevant tasks at [list of time.id]. Act: [type: 1, inter_obj_id: real int, inter_obj_name: xxx, inhand_obj_name: yyy, motion: free-form motion]
2. ...

**Feedback.**

Input:
1. Human intention at current time.
2. Current human tasks.
3. Robot-inferred tasks to enhance comfort with offered objects (if any).

You are the human. Decide if the robot's assistance align with your needs.

Instructions:
1. Assess if each robot task supports the human tasks and intention. The robot's task doesn't need to be an exact match but should be relevant in purpose, context, or object categories. Use common reasoning to decide if it helps meet your needs.
2. Consider each robot thought and object individually against the human tasks. Approve it if it meets any one of the human tasks; sequence does not matter.
3. Be fair in your judgment—avoid being too generous or too harsh.
4. Respond with yes/no for each, followed by an explanation. Ensure items are in a list.

Write in the following format. Do not output anything else:
Tasks: [yes, no, ...]
Reasons_tasks:
1. ...

## G   Prompt Details of the Assistive Agent

We show exact prompts for VLMs in building the assistive agent.

**Intention Discovery.**

Input:
1. Sequence of images showing human motion from your and human's perspectives.
2. Current time.
3. Inferred Big Five personality scores (ignore if empty—this means it's your first collaboration with this human).
4. Inferred human profile (ignore if empty—this means it's your first collaboration with this human).
5. Most relevant human intentions discovered at previous times (ignore if empty—this means it's the first intention of the day).
6. Most relevant human tasks discovered at previous times.ids (ignore if empty—this means it's the first task of the day).

You are a robot assisting a human. Identify 5 possible human intentions based on the current time and visual observations.

Instructions:
1. Map the observed human motion to 5 possible high-level intentions at the current time (without mentioning the specific motion).
2. Intention must align with human Big 5 scores and reflect all aspects of the profile, and be diverse yet reasonable based on the house layout and available objects.
3. Intention must be high-level and either human-centric (e.g., hygiene, sport, leisure) or room-centric (e.g., clean, organize, set-up). Do not mention specific objects.

4. Intention must have temporal dependence but be non-repetitive with the intentions and tasks at previous times in the input.

Write in the following format. Do not output anything else: Time: xxx am/pm
Intention 1: basic descriptions.
Reason_human: detailed descriptions of why it follows the Big 5 scores and profile.
Reason_intentions: detailed descriptions of why it has temporal dependence with the previous, relevant intentions at [list of time].
Reason_tasks: detailed descriptions of why it has temporal dependence with the previous, relevant tasks at [list of time.id].
Reason_vis: detailed descriptions with respect to the visual cues.
...

## Task Discovery.

Input:
1. Human intention at current time.
2. A dict mapping rigid, static furnitures to their IDs and rooms.
3. Inferred Big Five personality scores (ignore if empty—this means it's your first collaboration with this human).
4. Most relevant human intentions discovered at previous times (ignore if empty—this means it's the first intention of the day).
5. Most relevant human tasks discovered at previous times.ids (ignore if empty—this means it's the first task of the day).

You are a robot assisting a human.

Instructions:
1. Break down the intention into 5 tasks.
2. Task type: For each human task, provide one small, handable object from a magical box. Furnitures in the dict are for room understanding and cannot be used.
3. Tasks should be continuous and logical, and align with your Big 5 scores and profile.
4. Tasks must have temporal dependence with the intentions and tasks at previous times.
5. All objects are rigid and cannot deform, disassemble, or transform.

Write in the following format. Do not output anything else:
Time: xxx am/pm
Intention: basic descriptions.
Tasks:
1. Thought: detailed descriptions of the task. Reason_human: why it alignes with your Big 5 scores and profile. Reason_intentions: how it depends on previous, relevant intentions at [list of time]. Reason_tasks: how it depends on previous, relevant tasks at [list of time.id]. Act: [obj_name: xxx]
2. ...

## Traits Inference.

Input:
1. Human intentions at previous times (ignore if empty—this means it's your first inference).
2. Human tasks at previous times.ids.
3. Human profile (ignore if empty—this means it's your first inference).

Task: Mimic this human by:
1. Inferring Big Five personality traits (scale 1-5, float) based on the provided intentions and task.
2. Summarizing the human profile (i.e., preferences/habits) based on the intentions and tasks within three sentences. Revise the existing human profile if necessary.

Write in the following format. Do not output anything else:
Scores: {'openness': a, 'conscientiousness': b, 'extroversion': c, 'agreeableness': d, 'neuroticism': e}

```
Profile: ...
Reasons_ocean: explain each ocean.
Reasons_profile: explain the profile.
```

