# OpenReview forum: "COOPERA: Continual Open-Ended Human-Robot Assistance"
_NeurIPS.cc/2025/Conference — NeurIPS 2025 spotlight_

### Official Review · Reviewer_PjNb · 2025-06-29

**Clarity:** 3
**Significance:** 3
**Originality:** 3
**Rating:** 5
**Confidence:** 3

**Summary:**

This paper introduces COOPERA, a framework for continual, open-ended human-robot interaction with simulated humans driven by psychological traits and long-term intentions. The paper provides a new benchmark for such tasks, and presents a method to adapt to human behaviors by predicting their intentions.

**Questions:**

1. While the proposed method gains the best improvement over time, it seems that it does not converge to the one with ground truth intentions, as in Fig. 6. Will it help if you simulate for more days?
2. What is the baseline method used for comparison in Table 2?
3. How do you define success for each task? Any visualization/video on how the trained robot agent actually performs during testing?

**Ethical Concerns:**

["NO or VERY MINOR ethics concerns only"]

**Final Justification:**

My concerns are addressed and I raised my score in support of this paper.

**Quality:**

3

**Strengths And Weaknesses:**

Strength:
1. The long-term human-robot interaction problem is important and the benchmark provided along the paper can be useful.
2. Ablation experiment for the proposed method is comprehensive

Weakness:
1. The discussion on the benchmark problem is not sufficient, given it is a newly introduced and not a widely acknowledged testbed.
2. The comparison with baseline methods needs more discussions.

---

> ### Author Rebuttal · Authors · 2025-07-30
>
> Thank you for your insightful and constructive comments. We appreciate your recognition of the contribution of our framework and the introduced benchmark, as well as the design of our ablation study. This recognition is also kindly shared by other reviewers (Reviewer kYPv, Reviewer Zf7z, Reviewer irvc). In response to your suggestions, we conducted **additional experiment** to strengthen our analysis. For all HRI-related experiments, we report results from our main method under the more challenging collaboration type 2 (one trial per setting) due to time constraints. Below, we address each of your concerns.
>
> ---
>
> **Q1:** The discussion on the benchmark problem is not sufficient, given it is a newly introduced testbed.
>
> **A1:**
> Thank you for the valuable suggestions! We fully agree that providing additional explanations to Section 3.1 is essential, given that COOPERA is a newly introduced testbed. Central to COOPERA is our proposed human model and the continuous feedback mechanism operating at a daily scale. With these components, COOPERA differs from previous HRC benchmarks [1, 2] in two aspects:
>
> 1) Existing benchmarks typically focus on short-term, episodic HRC tasks that are closed-set and predefined. In contrast, our human model generates open-ended, environment-driven, and dynamic human activities that vary with time, enabling COOPERA to support long-horizon collaboration tasks spanning multiple days.
>
> 2) The daily feedback mechanism allows robots to continuously optimize their behavior over extended periods, leading to increasingly personalized interactions tailored to individual human traits and preferences.
>
> We will explicitly add these points to Section 3.1.
>
> ---
>
> **Q2:** The comparison with baseline methods needs more discussions.
>
> **A2:**
> Thank you for highlighting this! We strongly agree that additional detailed analysis on the performance of our method vs. the baselines are needed. For paper writing, we will include more analysis in Section 4.3 and move some of the implementation details in Section 4.1 to Appendix.
>
> Below we provide an in-depth discussion of why each method achieves distinct results. We evaluate COOPERA against six baselines: (1) Direct Prompting, (2) Direct Finetuning, (3) GT\_Intention, (4) Random, (5) Intention Agnostic, and (6) Human & Context Agnostic.
>
> For within-day improvement (Fig. 6a), our method achieves the highest performance gain by explicitly modeling the correlation between human intentions/tasks, traits, and temporal dependencies. As the day progresses, the robot accumulates interaction history and adapts its behavior to the human preferences. (1) Direct Prompting shows minimal improvement as it depends on retrieved interaction history without learning the human preferences. (2) Direct Finetuning performs slightly better but still struggles due to its rigid mapping from inputs to intentions/tasks, making it biased toward frequent training examples and overlooking the varying distribution of human behavior. (3) GT\_Intention, despite having ground-truth intentions, shows limited gains because receiving the correct intention upfront removes the need to model temporal context or adapt over time. Its performance is static by design and it serves as the upper bound method. (4) Random degrades over time due to the absence of validation. (5) Intention Agnostic shows moderate improvement but lags behind our method, as it skips intention inference and thus loses contextual information, which is problematic since temporal dependencies between intentions are typically stronger than between tasks (validated in Section 4.2). (6) Human & Context Agnostic captures only superficial time-task correlations and ignores human traits or temporal context, leading to weak within-day gains.
>
> Across-day improvement (Fig. 6b) further highlights the importance of learning the correlation between human intentions, human profile, and temporal dependencies. Our method achieves strong across-day gains—ranking second only to (3) GT\_Intention, which receives the correct daily intention and thus bypasses the most challenging aspect of the problem. (1) Direct Prompting, (2) Direct Finetuning, and (4) Random exhibit minimal across-day improvement for the same reasons they underperform within a day: lack of learning from feedback, rigid input-behavior mappings, or absence of validation. (5) Intention Agnostic and (6) Human & Context Agnostic show some improvement, but their performance remains significantly lower than our main approach, as they exclude critical components such as explicit intention modeling and the integration of human profile and temporal context.
>
> ---
>
> **Q3:** What is the baseline method used in Table 2?
>
> **A3:**
> Thank you for raising this point! We will add more explanations regarding the baselines for this experiment for better clarity. Table 2 evaluates our framework’s out-of-domain generalization (Section 4.3, line 319) in two scenarios: (1) scene generalization and (2) human generalization.
>
> In the scene generalization scenario, we first train the robot to collaborate with the same human across four distinct scenes (as defined in Setting 2: same human, different scenes). We then evaluate its ability to generalize to a fifth, unseen scene with the same human. The baseline here is the robot’s average performance during the initial interaction in the unseen scene without any finetuning on the previous four scenes, evaluated across 10 different humans.
>
> In the human generalization scenario, we train the robot to collaborate with three different humans in the same environment (Setting 3: different humans, same scene). We then assess its generalization to a fourth, previously unseen human. The baseline in this case is the robot’s initial, unadapted performance when interacting with the new human, without finetuning on the previous three, averaged across 5 new humans.
>
> Notably, our method outperforms the baseline in both cases, indicating that the robot learns transferable representations of human traits and environments. This shows that the system avoids overfitting and generalizes well to new humans and scenes.
>
> ---
>
> **Q4:** How do you define “success” for each task?
>
> **A4:**
> Thank you for pointing this out! We agree that evaluation clarity is important and will revise Section 4.3 (line 289) accordingly. In COOPERA, we assess F1-based success rates using three complementary methods, consistent with existing HRC benchmarks [1, 2, 3].
>
> In our framework, the robot-VLM predicts a superset of tasks to assist the human. An LLM-based classifier then filters this list, assigning "yes" or "no" to each task. For baselines that lack this component (e.g., Direct Prompting or Fine-tuning), we assume all predicted tasks are labeled "yes". Please see Section 3.3 for details.
>
> We measure success using three criteria:
>
> 1) Predicate-based success: Tasks are executed in the simulator and evaluated via predefined predicate functions (Appendix D.3, line 714). Matching is based on object classes, following Watch-and-Help [1], and F1 is computed over these outcomes.
>
> 2) LLM-based success: The simulated human (human-LLM) is shown the robot’s predicted tasks and the ground-truth intention, and gives binary ("yes/no") feedback on whether each task fulfills the intention (please see Appendix C.3 and F for detailed feedback machnism and exact prompt). F1 is computed from this feedback.
>
> 3) Human verification: Real humans replace the simulated human in (2) and provide judgments. This ensures our evaluation extends from simulation to real-world validation by human evaluators.
>
> ---
>
> **Q5:** Is there any visualization or video demonstration showing how the trained robot agent performs during testing?
>
> **A5:**
> Yes! For the predicate-based evaluation, the robot directly executes tasks in the simulated environment. We also provide functions for the human agent to perform actions collaboratively when the robot’s assistance is approved. This enables us to create visualizations and demonstration videos illustrating the human-robot collaboration process, as in Fig. 7 of our paper. Due to this year's conference policy restricting attachment of supplementary PDF during rebuttal, we are unable to include these visuals here. However, we will add additional visualization examples to the Appendix and our project website after the rebuttal period. Furthermore, we will release all code related to our framework.
>
> ---
>
> **Q6:** While the proposed method gains the best improvement over time, it seems that it does not converge to the one with ground truth intentions, as in Fig. 6.
>
> **A6:**
> This is a great question! We conduct one **additional experiment**. We extend the current setup by doubling the collaboration duration to observe continued adaptation trends.
>
> |Method|D1|D2|D3|D4|D5|D6|D7|D8|D9|D10|
> |-|:-:|:-:|:-:|:-:|:-:|:-:|:-:|:-:|:-:|:-:|
> |Main|0.236|0.250|0.342|0.406|0.431|0.425|0.455|0.473|0.478|0.470|
>
> From the results, we observe that performance stabilizes around day 8. While our benchmark focuses on collaboration up to day 5, our main interest lies in the personalization trend. Simulating longer HRC is an easy extension within our framework. We believe this stabilization reflects a natural saturation point. By day 8, most human preferences driven by traits and routines are learned, allowing the robot to form a reliable alignment. Given the capacity of our 7B model and LoRA-based finetuning, further gains become marginal without introducing more capable models or finer-grained learning signals such as multimodal cues or reasoning chains.
>
> ---
>
> **References:**
>
> [1] Watch-and-help: A challenge for social perception and human-ai collaboration. ICLR 2021.
>
> [2] Partnr: A benchmark for planning and reasoning in embodied multi-agent tasks. ICLR 2025.
>
> [3] NOPA: Neurally-guided Online Probabilistic Assistance for Building Socially Intelligent Home Assistants. ICRA 2023.

---

> > ### Comment · Reviewer_PjNb · 2025-08-05
> >
> > Thank you for providing more details on the benchmark problems and baseline methods. For the last point, I have a few follow-up questions.
> >
> > 1. The additional results you provided in the rebuttal do not seem to match well with the results in Fig. 6 in the original submission, any explanations? Are these results single trajectory or an average of multiple trajectories? In my opinion averaged results should be reported here.
> >
> > 2. As the additional results indicated, simulating for a longer time won’t help the proposed method to reach the performance of GTintent, and saturation occurs. In this case, I believe discussion of potential reasons is needed, also Fig. 6 (a) needs to be adjusted since GTintent has high performance at the beginning and it saturates right away, and comparing success rate increases against it only on day one is not totally fair.

---

> > > ### Author Response · Authors · 2025-08-05
> > > **Response to Reviewer PjNb**
> > >
> > > Thank you so much for your feedback. We are glad that earlier points are now clearer! Below we provide responses to your remaining concerns.
> > >
> > > ---
> > >
> > > **A1:** For the additional experiment, due to time constraints, we report the averaged results from our main method under the more challenging collaboration type 2, with one trial per setting. While this was briefly mentioned in the rebuttal, we understand it may not have been entirely clear. Thank you for carefully reading and engaging with the details!
> > >
> > > In contrast, the results shown in Fig. 6 of the paper represent averages across all collaboration types and settings, with multiple trials conducted per setting (please see Section 4.3, line 296 for details). This provides a more comprehensive evaluation for benchmark. As a result, there is a mismatch between the first-5 day results of our additional experiment and Fig. 6, since the former is a subset. We acknowledge this discrepancy and will complete the evaluation for consistency and integrate it into the paper with discussion.
> > >
> > > ---
> > >
> > > **A2:** This is a great point! It is true that GT_Intent still achieves higher performance compared to our main method after 10 days of collaboration. Our intention behind including GT_Intent in the benchmark is to provide an upper-bound reference, demonstrating performance when the robot always knows the ground-truth human intention. Receiving ground-truth intentions provides significant advantages, as accurately mapping visual observations of human performing detailed tasks to high-level intentions is a core challenge in HRC [1, 2], even in closed-set tasks. While our main method addresses this challenge in open-ended setting by integrating a robot-VLM and a finetuned intention LLM-classifier for improved alignment, directly receiving the ground-truth intention still offers a substantial edge.
> > >
> > > This advantage also explains why GT_Intent shows small improvement over days. Since GT_Intent bypasses the above challenge by receiving the correct intention upfront, it has less need to learn or leverage temporal dependencies between human intentions. This behavior is expected: such dependencies are often strong across hours (e.g., an intense workout at 11 am typically leads to lunch preparation at 12 pm), as validated in Section 4.2. As a result, GT_Intent’s performance remains static by design as its access to ground-truth intentions makes further adaptation unnecessary.
> > >
> > > Moreover, setting an upper-bound method is common practice in HRC benchmarks. For instance, Watch-and-Help [1] defines upper bound method, Oracle_B, where one agent always knows the ground-truth intention of the other agent. Our benchmark design is similarly inspired by prior work.
> > >
> > > That said, we fully agree more explanation is needed in our benchmark section (Section 4.3), and we will explicitly add this discussion. Additionally, we acknowledge Fig. 6 may currently cause confusion, and we intend to make the following changes: (1) In Fig. 6a, we will remove GT_Intent for fair comparison and explain it in the caption. (2) We plan to rename GT_Intent to a clearer term such as Oracle to explicitly indicate its role as an upper-bound reference method. We have made this adjustment ready in the paper for the next version.
> > >
> > > ---
> > >
> > > Please let us know if there are any other points we can clarify or discuss further. Once again, we greatly appreciate your insightful suggestions, which have strengthened our paper!
> > >
> > > ---
> > >
> > > **References:**
> > > [1] Watch-and-help: A challenge for social perception and human-ai collaboration. ICLR 2021.
> > >
> > > [2] NOPA: Neurally-guided Online Probabilistic Assistance for Building Socially Intelligent Home Assistants. ICRA 2023.

---

> > > > ### Comment · Reviewer_PjNb · 2025-08-06
> > > >
> > > > Thank you for the clarification. My concerns are addressed. I raised my score to 5 in support of this paper.

---

> > > > > ### Author Response · Authors · 2025-08-07
> > > > > **Response to Reviewer PjNb**
> > > > >
> > > > > Thank you so much for raising your score to Accept (5) and for the strong support of our paper! We sincerely appreciate your thoughtful discussion and valuable suggestions throughout the review process, which greatly improved the quality of our work.

---

### Official Review · Reviewer_irvc · 2025-06-30

**Clarity:** 3
**Significance:** 3
**Originality:** 3
**Rating:** 5
**Confidence:** 4

**Summary:**

This paper introduces COOPERA, a novel framework for continual, open-ended human-robot collaboration in simulated household environments. The core innovation lies in simulating psychologically grounded human agents using LLMs that generate temporally coherent, traits-driven behaviors across multiple days. The robot assistant learns from these interactions to infer human preferences and adapt its actions accordingly. The authors propose a benchmark, a method for human trait and task inference, and conduct thorough experiments on within-day and across-day adaptation, scene and human generalization, as well as human studies validating the realism of their simulations.

**Questions:**

1 All experiments use relatively small or mid-sized models (e.g., LLaMA-3-11B, Mistral-7B). Is there a particular computational or latency constraint driving this choice? Do the authors expect significantly better assistance behavior from larger models, or do smaller models already saturate performance given the task complexity?

2 How sensitive is the pipeline to the choice of LLM/VLM backbones? The paper includes one ablation replacing the robot’s VLM, but further experiments could highlight how brittle (or robust) the system is to different base models or prompting strategies.

3 While the study focuses on offline interactions and day-level adaptation, do the authors envision a version of COOPERA where real-time, in-episode adaptation (e.g., using continual feedback during a task) can be integrated? If so, what components would need to be modified?

4 Given the richness of the simulated human profiles and long-term interaction, are there any qualitative or longitudinal metrics that could better reflect improvement in human-robot support or perceived usefulness of the robot?

**Ethical Concerns:**

["NO or VERY MINOR ethics concerns only"]

**Final Justification:**

The issues are resolved and I updated my scores to accept.

**Limitations:**

Yes.

**Paper Formatting Concerns:**

NA.

**Quality:**

4

**Strengths And Weaknesses:**

Strengths:

The proposed setting fills a gap in HRC research by addressing long-horizon personalization and temporal reasoning. Before LLMs are widely studied, the cost of obtaining human data was high and it is useful now to have LLM included to simulate the data.

The paper provides diverse experiments, including simulation, user studies, and offline real-human tasks. The use of predicate-based, LLM-based, and human evaluations strengthens the validity of the findings.

The simulation of long-term human behavior driven by LLMs with embedded personality traits is original and adds realism to the evaluation framework.


Weaknesses:

While the study uses a reasonably sized model (e.g., Llama-3.2-11B), it remains unclear whether scaling up LLMs/VLMs (e.g., GPT-4, Gemini, Claude) would significantly enhance robot performance of the proposed methods or the baselines. The authors mention computational constraints but do not explore the trade-offs or potential gains. More experiments could validate if the performance gap between the proposed methods and baselines would become bigger or smaller with more advanced LLMs.

The method uses specific LLM prompts and retrieval heuristics; a deeper analysis of robustness across LLM backbones or prompt configurations would strengthen generalizability claims.

---

> ### Author Rebuttal · Authors · 2025-07-30
>
> Thank you for your insightful and constructive comments. We appreciate your recognition of the originality and the contribution of our framework, the novelty of our human simulation pipeline, and the comprehensiveness of our experimental design. This recognition is also kindly shared by other reviewers (Reviewer kYPv, Reviewer Zf7z, Reviewer PjNb). In response to your suggestions, we conduct **additional experiments** to strengthen our analysis. For all HRI-related experiments, we report results from our main method under the more challenging collaboration type 2 (one trial per setting) due to time constraints. Below, we address each of your concerns.
>
> ---
>
> **Q1:** How sensitive is the COOPERA pipeline to the choice of LLM/VLM backbone and how would models of different sizes impact the performance? What is the reason behind the current choice?
>
> **A1:**
> We choose LLaMA-3.2-11B as the robot-VLM and Mistral-7B-Instruct-v0.2 as the robot-LLM classifier for two reasons. First, both are open-source under research-friendly licenses, offering strong performance, transparency, and reproducibility for benchmarking. In contrast, proprietary models like GPT require costly API access, are frequently updated, and make results harder to reproduce. Second, 7B–11B models strike a practical balance between performance and computational efficiency [1, 2]. They support single-GPU inference (24GB RAM) and LoRA finetuning with just 2–3 GPUs, making them accessible to academic labs and the broader research community.
>
> Yet, we fully agree it is important to evaluate the framework’s robustness to different VLM backbones. We conduct two **additional experiments** by replacing the robot-VLM with a stronger model (GPT-4o) and a weaker one (Qwen2.5-VL-3B-Instruct).
>
> |Method|Day 1|Day 2|Day 3|Day 4|Day 5|
>  |-|:-:|:-:|:-:|:-:|:-:|
> |Qwen|0.196|0.218|0.295|0.364|0.373|
> |GPT|0.313|0.325|0.424|0.483|0.526|
>  |Main|0.236|0.250|0.342|0.406|0.431|
>
> As expected, stronger models produce better performance, aligning with findings in Embodied Agent Interface [3], a benchmark that tests the impact of LLM capacity on LLM-powered embodied AI systems. Notably, the performance gap between GPT-4o and LLaMA is larger than that between LLaMA and Qwen. However, smaller VLMs like Qwen still achieve reasonable performance at the end of day 5 and show a clear personalization trend. This demonstrates the robustness of our framework across varying model capacities.
>
> ---
>
> **Q2:** The method uses specific LLM prompts and parameters. Deeper analysis is needed.
>
> **A2:**
> Thank you for pointing this out! We agree that prompt design and parameter tuning are important considerations for any LLM-based system. To maintain full control and interpretability of our framework, we handwrote our prompts. However, we also acknowledge there are open-source tools that automate prompt optimization. To ensure our performance is not the result of over-engineered prompts, we conduct an **additional experiment** by integrating DSPy [4, 5], an open-source framework for self-composing and self-improving prompts, into our assistive robot agent. DSPy allows prompt tuning using only high-level instructions and input-output examples. We evaluate our system with DSPy-generated prompts.
>
> |Method|Day 1|Day 2|Day 3|Day 4|Day 5|
> |-|:-:|:-:|:-:|:-:|:-:|
> |DSPy|0.222|0.262|0.355|0.387|0.439|
> |Main|0.236|0.250|0.342|0.406|0.431|
>
> We observe a slight performance gain using DSPy. DSPy leverages the OPRO algorithm [8], and its evaluation reports an average gain of 5% in expert domains. In our case, we see a 3% improvement, which we find reasonable. It suggests that our hand-written prompts are already well-designed, though not overly engineered.
>
> ---
>
> **Q3:** Can COOPERA be extended to support real-time, in-episode adaptation (e.g., continual feedback during a task)? What components of the current system would need to be modified to support this?
>
> **A3:**
> This is a great question! COOPERA currently focuses on day-level adaptation, which we believe better reflects real-world usage—robot is optimized overnight while humans rest. However, enabling real-time adaptation through hour-level human/between-task feedback and developing corresponding online learning algorithms is a promising direction and a fundamental challenge for any LLM-based agent.
>
> To conduct a preliminary study for this extension, we conduct one **additional experiment**.
>
> COOPERA with hour-based communication: The human’s hourly intention/task is influenced by the robot’s action in the previous hour. For example, if the robot fails to clean the living room, the human may skip yoga to finish cleaning. We modify the human prompt to include robot's success/failure. We also ask the human to provide hourly feedback with reasoning to the robot (e.g., I disapprove this task because...). This helps the robot understand its mistakes and adjust its next action accordingly in the next hour.
>
> |Method|Day 1|Day 2|Day 3|Day 4|Day 5|
> |-|:-:|:-:|:-:|:-:|:-:|
> |Hour-Based|0.224|0.243|0.323|0.401|0.415|
> |Main|0.236|0.250|0.342|0.406|0.431|
>
> The results are interesting. We expect higher or comparable performance due to frequent feedback but observe a slight drop. We find that the robot often repeats the previous intention to “make amends” when the human disapproves, while the simulated human sticks to its traits and only notes the error in feedback. Extending COOPERA to support more frequent feedback and developing new assistive strategies are promising directions for future work.
>
> ---
>
> **Q4:** Given the long-term nature of the simulation and human profiles, are there qualitative or longitudinal metrics that could better capture: 1) The robot’s improvement over time? 2) The perceived usefulness or support quality from the human’s perspective?
>
> **A4:**
> Thank you for the suggestion! Our current evaluation focuses on quantitative performance (predicate-based, LLM-based, and human-verified success rates), following conventions in existing HRC benchmarks [6, 7]. We fully agree that qualitative and longitudinal metrics can better capture perceived usefulness and long-term improvement.
>
> We propose two directions for future work:
>
> 1) Subset-Based Evaluation for Intention Alignment Over Time: At each hour, the robot infers the human intention and proposes a task set. Our current LLM-based evaluation only gives binary approval/disapproval to each task in the set. We can extend this by selecting subsets of the robot’s proposed tasks and asking the simulated human whether each subset fulfills their intention. Ideally, some subsets will be approved, others rejected. By recursively forming and evaluating the subsets, we can determine which combinations of tasks are most critical for intention fulfillment. We can then assign a final score to the original full set by summing the scores of approved subsets (+1 each) and disapproved subsets (-1 each).
>
> 2) Human Preference Between Successive Robot Policies: For HRC at day t = n, we can compare a robot finetuned on data from t = 0 to n-1 versus t = 0 to n-2, then ask the simulated human to express a preference. This offers insight into how alignment improves across days.
>
> ---
>
> **References:**
>
> [1] Grattafiori, Aaron, et al. The Llama 3 Herd of Models. ArXiv, 2024.
>
> [2] Kaplan, Jared, et al. Scaling Laws for Neural Language Models. ArXiv, 2020.
>
> [3] Li, Manling, et al. Embodied Agent Interface: Benchmarking LLMs for Embodied Decision Making. NeurIPS, 2024.
>
> [4] Khattab, Omar, et al. DSPy: Compiling Declarative Language Model Calls into Self-Improving Pipelines. ArXiv, 2023.
>
> [5]  Xian, Jasper, et al. Prompts as Auto-Optimized Training Hyperparameters: Training Best-in-Class IR Models from Scratch with 10 Gold Labels. ArXiv, 2024.
>
> [6] Puig, Xavier, et al. Watch-and-help: A challenge for social perception and human-ai collaboration. ICLR, 2021.
>
> [7] Chang, Matthew, et al. Partnr: A benchmark for planning and reasoning in embodied multi-agent tasks. ICLR, 2025.
>
> [8] Yang, Chengrun, et al. Large language models as optimizers. ICLR, 2024.

---

> > ### Comment · Reviewer_irvc · 2025-08-03
> >
> > Thank you for the detailed response and additional experiments. They help clarify several points I raised.
> >
> > Q1 (Model size):
> > The comparison across GPT-4o, LLaMA, and Qwen was helpful. It’s good to see the framework performs reasonably even with smaller models, while stronger ones bring improvements.
> >
> > Q2 (Prompt robustness):
> > The DSPy results suggest the prompts are already well-designed, and that performance isn’t overly dependent on manual tuning, which is reassuring.
> >
> > Q3 (Real-time adaptation):
> > The hour-level experiment is an interesting step toward more interactive learning. I agree it's a hard problem and worth future exploration, outside the scope of this work.
> >
> > Q4 (Long-term metrics):
> > Your proposed evaluation ideas sound promising, especially the subset-based scoring and policy comparisons. They could offer more insight into long-term alignment.
> >
> > Overall, I found the response convincing and clear. I am updating my overall score to accept based on improved clarity and stronger support for the contributions.

---

> > > ### Author Response · Authors · 2025-08-08
> > > **Response and Thank Note to Reviewer irvc for Final Acknowledgment and Justification**
> > >
> > > Dear Reviewer,
> > >
> > > Thank you once again for the valuable discussion and for raising your score in support of our work! Your insightful suggestions have greatly improved its quality. We will integrate the additional experiments (Q1–Q3) either into the ablation study (if space permits) or into the Appendix with clear references.
> > >
> > > As we are near the end of the discussion period, please don’t hesitate to let us know if we can provide any further clarification to support your final acknowledgment and justification. (As a kind note, this year’s NeurIPS system requires reviewers to click the Mandatory Acknowledgment button and provide a short final justification in the rebuttal interface). Thank you for your time and thoughtful engagement throughout this process!

---

> ### Author Response · Authors · 2025-08-03
> **Response to Reviewer irvc**
>
> We are happy our replies addressed your questions! Thank you for your kind response and for improving the score from 4 (Weak Accept) to 5 (Accept) in support of our work. We will add the additional experiments to the paper or Appendix with clear references.

---

### Official Review · Reviewer_Zf7z · 2025-06-30

**Clarity:** 2
**Significance:** 3
**Originality:** 3
**Rating:** 4
**Confidence:** 3

**Summary:**

The paper introduces COOPERA, a framework designed for continual, open-ended human-robot collaboration. It simulates human behaviors across multiple days using LLMs, driven by psychological traits and long-term intentions. A robot assistant agent leverages a VLM and another LLM to infer human intentions, classify tasks, and offer collaborative actions. Evaluations are conducted within simulated environments, assessing the robot’s ability to personalize assistance by learning human traits and adapting to context-dependent intents.

**Questions:**

1. In Figure 6b, intent-agnostic and human-agnostic models show comparable learning progress to the full proposed approach. Does this indicate that the system learns general collaboration strategies rather than specific personalized intent inference?
2. Clarify how feedback was provided to the robot at the end of each day. Was it binary (e.g., correct/incorrect predictions), detailed action-level feedback, or summary-level feedback?
3. How exactly were actions presented to the robot? Did the robot rely purely on visual observations, or was textual context also provided? Given the complexity of certain tasks described (e.g., brainstorming), clarification on inference challenges from visual data alone is warranted.
4. Clearly define the criteria for determining success in collaborative tasks. Were these predefined success conditions, user-defined outcomes, or evaluated through additional automated metrics?

**Ethical Concerns:**

["NO or VERY MINOR ethics concerns only"]

**Final Justification:**

The authors' detailed rebuttal and additional experiments addressed concerns regarding evaluation clarity and robustness across foundation models. While the realism and validity of synthetic human simulations remain partially unresolved, I acknowledge that these limitations are inherently challenging. Structural clarity and scope also remain ambitious and could benefit from further refinement. The framework and assistance methodology could benefit from further restructuring, potentially as separate papers or an expanded journal article. Given the improvements, my recommendation moved from Borderline Reject (3) to Borderline Accept (4).

**Limitations:**

While the authors acknowledge the difficulty in validating simulated human behaviors, explicitly addressing the risk of "echo chamber" effects due to reliance on LLM-generated data would strengthen the paper's transparency.

**Paper Formatting Concerns:**

No major formatting issues.

**Quality:**

3

**Strengths And Weaknesses:**

Strengths:
The work addresses a clearly motivated and significant problem and it provides thorough documentation of experimental settings, parameters, and model specifics. The analysis conducted evaluates multiple dimensions of the proposed approach, offering insights into the effectiveness and limitations of the system. The integration and application of foundation models in HRC tasks demonstrates thoughtful methodological design, leveraging existing technology to advance the field.

Weaknesses:
The evaluation heavily depends on synthetic human simulations, with limited validation against genuine human behaviors, raising questions about real-world applicability. The methods for assessing the realism of simulated humans are restricted primarily to classifier-based distinguishability and basic user studies, lacking more comprehensive measures of realism or plausibility. Additionally, the human studies narrowly focus on personality type identification and task identification, neglecting the broader context of expected or realistic behavior. There is a notable dependence on foundation models throughout the entire pipeline, which raises concerns regarding the genuine methodological novelty. The dense structuring of the content limits the nuanced discussion of results and findings and the paper appears to insufficiently explore or discuss potential system failures, both in terms of human simulation realism and robot assistance effectiveness.

Other comments:
Figure 1 appears not to be referenced in the text. Including explicit references or discussing it in relation to the framework's overview would enhance clarity.

Given the complexity and ambitious scope of the proposed framework, the content seems more suitable for a journal article where the authors can extensively discuss each component with greater depth. Alternatively, the paper might benefit from clearly delineating the presentation into separate components: one part thoroughly evaluating the COOPERA framework's capability to simulate human behavior realistically, and another part focused on the robot assistance approach. As currently structured, it is challenging to fully trust the assistance methodology without first establishing comprehensive confidence in the fidelity and realism of the underlying COOPERA simulation framework.

Quality: Technically solid, though reliance on synthetic data without substantial real-world validation limits completeness.
Clarity: Generally clear but overly dense; improved organization or division into separate works would greatly enhance readability.
Significance: Addresses an important, challenging HRC problem with potential high impact, but practical significance uncertain due to limited realism validation.
Originality: Novel integration of existing foundation models in a comprehensive continual collaboration framework, clearly differentiating from prior methods.

---

> ### Author Rebuttal · Authors · 2025-07-30
>
> Thank you for your insightful and constructive comments. We appreciate your recognition of our framework’s contribution, careful integration of foundation models, and completeness of our evaluation. This recognition is also kindly shared by other reviewers (kYPv, irvc, PjNb). Following your suggestions, we added **new experiments** to strengthen our analysis. For HRI experiments, we report results under the more challenging collaboration type 2 (one trial per setting) due to time constraints. Below, we address each of your concerns.
>
> ---
>
> **Q1:** The evaluation relies heavily on synthetic human simulations, with limited validation against real human behavior.
>
> **A1**: Thank you for pointing this out! COOPERA relies on synthetic human simulations due to ethical, safety, and practical constraints. Our goal is to study long-term, open-ended HRC, which better mirrors real-world use, unlike most HRC benchmarks focused on episodic, closed-set tasks [1–3]. To support long-horizon HRC, we need humans who can continuously collaborate with robots and provide feedback. However, running full real-world studies with physical robots in household over multiple days is costly and raises ethical and safety concerns. As an initial step, we build a scalable simulation pipeline enabling personalized, long-term HRC entirely in simulation (supported by Reviewer kYPv, irvc).
>
> To assess the realism of our simulated human model, we present extensive analysis in Section 4.2 (also noted by Reviewer irvc), including classifier-based evaluations (distinguishability, diversity, psychometric coherence, temporal dependence) and two user studies. However, we fully agree more validation is necessary.
>
> To address your concern, we conduct a **new experiment**: six users provide traits and daily intentions over 5 days. Using the same traits, we prompt the LLM to simulate intentions for the same period. Both sets are aggregated into single paragraphs (removing time formatting like “9am:” to avoid inflated similarity), and semantic similarity is computed using SBERT (all-mpnet-base-v2) and OpenAI embeddings (text-embedding-3-small). We compare against three baselines: (1) prompting without human profile (generic), (2) mismatched LLM-human intention pairs, and (3) generating all intentions at once without hour-level grounding (as in Table 4 ablation).
>
> Method|SBERT|OpenAI Embed
> -|:-:|:-:
> Generic|0.554|0.537
> Misalign|0.523|0.543
> All-Day Intent|0.769|0.746
> Main|0.810|0.772
>
> From the results, Generic and Misaligned yield moderate similarity ($\sim$0.5), since sentence encoders naturally assign partial similarity to structurally similar content. Our main method achieves notably higher scores, indicating closer alignment with human-specific intent. Sentence-BERT yields slightly higher similarity than OpenAI embeddings, likely due to its objective being explicitly tuned for sentence-level semantic comparisons.
>
> ---
>
> **Q2:** Notable dependence on foundation models throughout the pipeline.
>
> **A2:** Great point! Both our human simulation pipeline and benchmark methods rely on foundation models. As COOPERA extends closed-set tasks [1, 3] to open-ended ones, foundation models are a natural choice for agent reasoning and decision-making [3], a trend also seen in open-ended embodied AI systems [4].
>
> Yet, we fully agree on concerns with heavy reliance on LLMs, especially regarding pipeline brittleness/robustness and prompt over-engineering.
>
> To address the concerns, we run three **new experiments**: 1) We replace the robot-VLM with a stronger model (GPT-4o) and 2) a weaker one (Qwen2.5-VL-3B-Instruct). 3) Test prompt sensitivity by swapping our handwritten prompts with DSPy [5], an open-source framework for prompt automation using only high-level instructions and I/O examples.
>
> Method|Day 1|Day 2|Day 3|Day 4|Day 5
> -|:-:|:-:|:-:|:-:|:-:
> Qwen|0.196|0.218|0.295|0.364|0.373
> GPT|0.313|0.325|0.424|0.483|0.526
> DSPy|0.222|0.262|0.355|0.387|0.439
> Main|0.236|0.250|0.342|0.406|0.431
>
> As expected, stronger models yield better performance [3]. Notably, smaller VLMs like Qwen still achieve reasonable performance by day 5 and show a clear personalization trend, demonstrating the robustness of our framework across model scales. We also observe a 3% gain with DSPy, compared to its reported result (5%) in expert domains. This suggests our handwritten prompts are well-structured but not over-engineered.
>
> ---
>
> **Q3:** The use of synthetic data without substantial real-world validation raises concerns about completeness and the risk of “echo chamber” effects. Real-world applicability remains unclear.
>
> **A3:** Thank you for bringing this up! The ultimate goal of COOPERA is to develop robot agents that assist real humans by adapting to their preferences over time. To validate real-world applicability (Section 4.4), we conducted human verification and collaborated with offline real humans. While our experiments are in simulation, the robot’s embodiment and actions match the real Fetch robot’s default settings in Habitat [7], enabling seamless transfer of generated action sequences to the real robot.
>
> However, we fully agree more validation of real-world applicability is necessary. To address this, we include a **new experiment**. Three real humans replace the LLM and collaborate under setting 2. Users propose intentions based on their traits and adjust to robot predictions in real time. Users are shown retrieved object/motion sets and choose which to interact with and use. For simplicity and due to time constraints, users input responses as text instead of using keyboard to control the simulated agent.
>
> Method|Day 1|Day 2|Day 3|Day 4|Day 5
> -|:-:|:-:|:-:|:-:|:-:
> HITL|0.279|0.340|0.411|0.393|0.457
> Main|0.290|0.324|0.371|0.476|0.440
>
> The results are interesting. We expect lower performance due to real human variability, yet results are higher. After discussing with participants, we find they tend to recall and follow their mid-week routines, which are relatively stable. This consistency leads to lower variability than our simulated humans, allowing the robot to personalize more quickly and validating the behavioral diversity in our simulated humans.
>
> ---
>
> **Q4 & 5:** Clarification on how success is defined and how feedback is provided to the robot.
>
> **A4 & 5:** Thank you for pointing this out! Evaluation clarity is crucial and we will revise Section 4.3. We compute F1-based success using three methods, consistent with prior HRC benchmarks [1–3]. The robot’s LLM classifier outputs a yes/no-labeled task list, and F1 is computed using the following GT labels:
>
> 1) Predicate-based: Tasks are executed and evaluated by predicate functions (Appendix D.3), with class-based object matching following Watch-and-Help [1].
>
> 2) LLM-based: Simulated human judges robot tasks and gives binary feedback on if each task fullfills its intention (Appendix C.3 and F).
>
> 3) Human verification: Same as (2) but real humans provide the binary feedback.
>
> For more details, please see our reply to Reviewer PjNb Q4 due to word limit.
>
> ---
>
> **Q6:** In Fig. 6b, intent-agnostic and human-agnostic baselines show comparable learning progress to full proposed approach.
>
> **A6:**
> Great question. Among six baselines (Section 4.3), Intention Agnostic is closest to our full method. It removes intention inference so tasks are predicted without modeling intention. Human & Context Agnostic learns only time-to-task mappings, ignoring traits and temporal cues.
>
> In Fig. 6b, Intention Agnostic and Human & Context Agnostic reach $\sim$0.4 and $\sim$0.35 F1, while our full method achieves $\sim$0.5. Though the gap seems small due to averaging across all settings, Appendix Fig. 12 reveals: (1) Our method significantly outperforms both in harder settings (3 & 4, different humans), highlighting the value of intention modeling and personalization. (2) In easier settings (1 & 2, same human), both baselines perform similarly since trait learning matters less. (3) In multi-human settings (3 & 4), Intention Agnostic surpasses Human & Context Agnostic, showing while intention modeling helps, learning human-specific behavior is crucial for generalization.
>
> ---
>
> **Q7:** How are robot actions defined, and what input modalities support complex task inference?
>
> **A7:**
> Robot actions are implemented as primitives from Habitat’s task\_action registry, combining navigation EE control: (1) MoveEEAction moves the EE via Cartesian steps with IK; (2) PickObjIdAction grasps objects by snap within a threshold; (3) PlaceObjIdAction releases via desnap; (4) ResetEEAction resets the EE to default. Navigation uses classical planning with AABB thresholds. These form a full pick-and-place pipeline. We will release all code.
>
> For intention discovery, the robot-VLM receives five 1024×768 RGB frames of the human’s first task in collab type 2. In type 1, we also provide texts (Section 3.1). Intention-to-task decomposition is text-based. For complex intentions (e.g., “meditate”), we let VLM reason over visual cues, prior actions, and traits. For complex tasks (e.g., “brainstorm”), we use the VLM’s proposed task superset, filtered by our LLM-based classifier, to capture valid actions.
>
> ---
>
> **Q8:** Paper writing. The structure limits the nuanced discussion of results and findings. Fig. 1 not referenced.
>
> **A8:** Thanks for pointing out! For revision, we will: (1) move selected implementations from Sections 3.2 and 4.2 to Appendix; (2) expand discussion of findings, including new results addressing Q1–Q3 and Q6; and (3) clarify how success rate and feedback are defined, either in the main text or via Appendix references. We will also explicitly describe Fig. 1 in intro.
>
> ---
> [1] Watch-and-help. https://arxiv.org/abs/2010.09890
>
> [2] Partnr. https://arxiv.org/abs/2411.00081
>
> [3] NOPA. https://arxiv.org/abs/2301.05223
>
> [4] Embodied Agent Interface. https://arxiv.org/abs/2410.07166
>
> [5] DSPy. https://arxiv.org/abs/2310.03714
>
> [6] Habitat 2.0. https://arxiv.org/abs/2106.14405

---

> > ### Comment · Reviewer_Zf7z · 2025-08-06
> >
> > Thank you for your detailed responses and the additional experiments. Your efforts have helped address several of my earlier questions. While these additions are valuable and clarify some aspects, I remain somewhat skeptical about the overall validity and realism of the simulated behaviors. Although I recognize the practical challenges involved, I still have reservations about fully trusting the COOPERA framework, given the heavy reliance on synthetic data and simulations. Admittedly, it's not entirely clear what level or type of validation would fully alleviate these concerns.
> >
> > Nonetheless, given the thoroughness of your response and the new analyses provided, I believe the paper has improved. Thus, I'm adjusting my recommendation to Borderline Accept, while noting that the ambitious scope and underlying validity remain areas of cautious consideration.

---

> > > ### Author Response · Authors · 2025-08-06
> > > **Response to Reviewer Zf7z**
> > >
> > > We are happy our replies addressed your questions! And we sincerely appreciate your decision to improve the score from 3 (Borderline Reject) to 4 (Borderline Accept). The improved quality of the paper would not have been possible without your valuable and insightful suggestions.
> > >
> > > Meanwhile, we fully understand your remaining reservation regarding the validity of simulation. We will take the following immediate steps and also look forward to exploring the following directions in the future:
> > >
> > > 1) We will integrate the additional experiments discussed in Q1 and Q3 into Sections 4.2 and 4.4 of the paper. For the one in Q2, we will include it either as an ablation (if space permits) or in the Appendix with clearer references.
> > >
> > > 2) We will move selected implementation details from Sections 3.2 and 4.2 to the Appendix, and add more detailed discussion regarding feedback mechanism, success rates, and benchmark results.
> > >
> > > 3) Migrating COOPERA to a real-world setting is definitely both challenging and promising. Topics such as seamless transfer, setting up and studying safe long-horizon HRC in the real world, and improving human modeling and validation are all important. We plan to actively pursue these directions with COOPERA as a foundation.

---

### Official Review · Reviewer_kYPv · 2025-07-02

**Clarity:** 2
**Significance:** 3
**Originality:** 3
**Rating:** 4
**Confidence:** 4

**Summary:**

This paper presents a novel framework for human-robot interaction (HRI) with language models as simulated human agents. The authors performed standard evaluation on both the framework itself and a curated robotic VLM agent. This novel task opens up new challenges and opportunities in embodied research.

**Questions:**

1. Why is a fixed daily routine selected instead of leaving room for interaction?
2. What is the performance of the baselines after sufficient days? If the generated data has limited days / episodes, maybe try repeating the overall routine as a cycle.
3. From the provided Figure 12 in Appendix, I see clear patterns between each day, i.e., increase->decrease within one day and increase even more on the second day. Is it coming from the task setting or the agent paradigm?

I will consider increasing my score if the authors provide additional details to address my concerns.

**Ethical Concerns:**

["NO or VERY MINOR ethics concerns only"]

**Final Justification:**

The authors addressed my concerns properly. The unusual results of human-in-the-loop and increased HRI raises new points worth of discussion which I'm looking to see in the final version should the chair decided to accept this paper. That apart, I still recommend this paper but certain drawbacks including the soundness of the task itself and the quality of the dataset refrains me from giving a score of 5.

**Limitations:**

yes

**Quality:**

3

**Strengths And Weaknesses:**

Strengths:
The idea of simulating human behavior through the profile -> intent -> task -> execution pipeline is quite interesting. To me, it is even worthier of discussion than the overall framework. Nonetheless, the framework provides a novel perspective of HRI and opens up possibilities for large scale data collection and training.

Weaknesses:
1. The human agent follows a pre-set routine of actions, but normally a human would at least adapt to what the robot is doing, etc. clearing the path for the robot.
2. The experiments focus on the success rate increase over time, which is kind of a test-time learning setting. I would expect more analysis on the 'final success rate', the success rate after sufficiently days / episodes so that in theory the agent's intention is deducible from its actions.

---

> ### Author Rebuttal · Authors · 2025-07-30
>
> Thank you for your insightful and constructive comments. We appreciate your recognition of the novelty and contribution of our framework, and your interest in our human simulation pipeline. This recognition is also kindly shared by other reviewers (Zf7z, irvc, PjNb). Following your suggestions, we conduct **additional experiments** to strengthen the analysis. For all HRI-related experiments, we report results from our main method under the more challenging collaboration type 2 (one trial per setting) due to time constraints. Below, we address each of your concerns.
>
> ---
>
> **Q1:** Why is a fixed daily routine selected for the simulated human agent instead of enabling interaction to the robot?
>
> **A1:**
> Thank you for pointing this out! We believe “fixed vs. dynamic” routines can be interpreted in two ways. The first is if a human with specific traits shows varied daily behavior while following a general routine (e.g., Monday 9am for cleaning, Tuesday 9am for exercise). We achieve this by setting a high LLM temperature (line 171). The second concerns if the human adjusts plans based on external factors such as mood, weather, or the robot’s behavior.
>
> While our framework shows robustness to temporal factors by offline real human collaboration (Section 4.4, Table 3), we did not explicitly simulate human adaptation to robot behavior. Enabling reactive human behavior requires a cognitively richer agent and defining clear criteria for when and how adaptation occurs. This introduces additional complexity, turning the setup into a multi-agent framework. Our current goal is to develop an assistive robot that can improve across long-horizon, and we view simulating mutual adaptation as promising future work.
>
> Yet, we fully agree it is valuable to explore human plan adaptation in response to robot behavior. We conduct two **experiments** to study this extension.
>
> 1) Increased human-robot interaction (Increased HRI): The human’s hourly intention/task is influenced by the robot’s action in the previous hour. For example, if the robot fails to clean the living room, the human may skip yoga to finish cleaning. We modify the human prompt to include robot's success/failure. We also ask the human to provide hourly feedback with reasoning to the robot (e.g., I disapprove this task because...). This helps the robot understand its mistakes and adjust its next action accordingly in the next hour.
>
> 2) Human-in-the-Loop (HITL): Three real humans replace the LLM and collaborate under setting 2. Users propose intentions based on their traits and adjust to robot predictions in real time. Users are shown retrieved object/motion sets and choose which to interact with and use. For simplicity and due to time constraints, users input responses as text instead of using keyboard to control the simulated agent.
>
> |Method|D1|D2|D3|D4|D5|
> |-|:-:|:-:|:-:|:-:|:-:|
> |Increased HRI|0.224|0.243|0.323|0.401|0.415|
> |Main|0.236|0.250|0.342|0.406|0.431|
>
> |Method|D1|D2|D3|D4|D5|
> |-|:-:|:-:|:-:|:-:|:-:|
> |HITL|0.279|0.340|0.411|0.393|0.457|
> |Main|0.290|0.324|0.371|0.476|0.440|
>
> The results are interesting. For Increased HRI, we expect higher or comparable performance due to frequent feedback but observe a slight drop. We find that the robot often repeats the previous intention to “make amends” when the human disapproves, while the simulated human sticks to its traits and only notes the error in feedback. For HITL, we expect lower performance due to real human variability, yet results are higher. After discussing with participants, we find they tend to recall and follow their mid-week routines, which are relatively stable. This consistency leads to lower variability than our simulated humans, allowing the robot to personalize more quickly and validating the behavioral diversity in our simulated humans.
>
> ---
>
> **Q2:** What is the performance of the baseline(s) after sufficient days (in contrast to test-time learning setting)?
>
> **A2:**
> Good question! To clarify, our framework includes two layers of test-time learning. The first layer operates within a day by prompting the robot-VLM with retrieved interaction history and the inferred human profile. The second layer operates across days by finetuning the robot-LLM classifier at the end of each day using data labeled from human feedback over previous days. For HRI at day t = n, the robot trains on data from days 0 to n–1, resulting in progressively more personalized behavior.
>
> We choose this test-time learning setup for two reasons. First, within-day adaptation improves both success rate throughout a day and final success rate, as in Fig. 6a and Table 5. Second, across-day adaptation reflects a realistic deployment scenario: when a user purchases a home assistant or cleaning robot, they should be able to deploy it immediately, and the robot should improve over time through daily interactions. This contrasts with prior HRI setups that rely on collecting all data upfront and training the model only once [1, 2].
>
> Yet, we fully agree it is valuable to study the train-all-at-once  setting. We conduct two **additional experiments**. (1) We extend the benchmark by doubling the collaboration duration to observe continued adaptation trends. (2) we remove across-day learning and train the robot once after the final day, then evaluate its performance on all days.
>
> |Method|D1|D2|D3|D4|D5|D6|D7|D8|D9|D10|
> |-|:-:|:-:|:-:|:-:|:-:|:-:|:-:|:-:|:-:|:-:|
> |Main (test-time)|0.236|0.250|0.342|0.406|0.431|0.425|0.455|0.473|0.478|0.470|
> |Train-All-at-Once |0.509|0.472|0.474|0.458|0.516|0.421|0.487|0.467|0.491|0.446|
>
> For the test-time setting, performance stabilizes around day 8. While our benchmark uses 5-day collaboration, our main interest lies in the personalization trend, and longer HRC is easily supported. By day 8, the robot has learned most human preferences shaped by traits and routines, forming reliable alignment. Given the 7B model with LoRA finetuning, further gains become marginal without more capable models or finer-grained signals like multimodal cues or reasoning chains.
>
> For the train-all-at-once case, daily performance is slightly higher or on par with the final days of test-time learning. This is because the robot remains unoptimized across all days, leading the simulated humans to disapprove more tasks. These disapprovals generate more negative samples for training. However, this comes at the cost of lacking any further personalization after the initial collaboration on day 1.
>
> ---
>
> **Q3:** In Fig. 12, there is a pattern across days (increase → decrease within a day, higher increase on second day). Is this pattern due to your task setting or the learning paradigm of the agent?
>
> **A3:**
> This is a great question! From Fig. 12, we observe performances tend to decrease around midday and recover toward the end of the day. We believe this pattern is due to two reasons.
>
> First, it reflects how human routines are structured both in reality and in our simulation. Generally, humans tend to engage in more predictable activities in the morning and evening (e.g., hygiene, eating, relaxing). These behaviors are easier for the robot to predict and assist. In contrast, midday behavior tends to be more diverse and strongly aligned with human traits. Since our human simulation pipeline samples intentions/tasks based on both traits and time, this increases the diversity of midday intentions/tasks, making inference and alignment harder for the robot. As a result, performance temporarily drops. The performance recovers as human routines re-stabilize in the evening or next morning. Importantly, despite these fluctuations, performance improves overall across days.
>
> Second, from Fig. 12, this fluctuation is amplified in more challenging task settings defined in our framework (Section 3.1). In Setting 1 (same human, same scene), the robot benefits from consistent exposure to the same human and environment, so midday dips are relatively small. In Setting 2 (same human, different scenes), unfamiliar object layouts introduce grounding challenges that increase midday variability. In Setting 3 (different humans, same scene), rotating between humans interrupts personalization, making intention interpretation harder, especially around midday when behavior is less routine. Finally, Setting 4 (different humans, different scenes) combines both factors, resulting in the largest fluctuations as the robot must generalize across both human and spatial contexts.
>
> To further validate this pattern, we conduct two **additional experiments** by replacing the robot-VLM with a stronger model (GPT-4o) and a weaker one (Qwen2.5-VL-3B-Instruct). To quantify temporal fluctuation independently of overall performance trend, we compute the Coefficient of Variation of Differences (CVD), defined as the standard deviation of hour-to-hour differences normalized by the mean performance (in %).
>
> **Success Rate**
> |Method|D1|D2|D3|D4|D5|
> |-|:-:|:-:|:-:|:-:|:-:|
> |Qwen|0.196|0.218|0.295|0.364|0.373|
> |GPT|0.313|0.325|0.424|0.483|0.526|
> |Main|0.236|0.250|0.342|0.406|0.431|
>
> **CVD**
> |Method|D1|D2|D3|D4|D5|
> |-|:-:|:-:|:-:|:-:|:-:|
> |Qwen|8.71|7.70|7.18|7.41|5.23|
> |GPT|6.28|5.91|4.20|4.32|2.35|
> |Main|8.46|7.06|7.49|6.21|3.77|
>
> As expected, stronger models produce better performance [3]. Notably, (1) the performance gap between GPT and llama is greater than between llama and Qwen. (2) while all VLMs exhibit fluctuation, stronger models like GPT show lower CVD. We attribute this to stronger models not only achieve better visual grounding for intention prediction but also generate more coherent intention-task proposals across time by better understanding prior interaction history.
>
> ---
>
> [1] Watch-and-help: A challenge for social perception and human-ai collaboration. ICLR 2021.
>
> [2] Partnr: A benchmark for planning and reasoning in embodied multi-agent tasks. ICLR 2025.
>
> [3] Embodied Agent Interface: Benchmarking LLMs for Embodied Decision Making. NeurIPS 2024.

---

> > ### Comment · Reviewer_kYPv · 2025-08-05
> >
> > Thanks to the authors for the prompt response to my questions! The unusual results of human-in-the-loop and increased HRI raises new points worth of discussion which I'm looking to see in the final version should the chair decided to accept this paper. That apart, I still recommend this paper but certain drawbacks including the soundness of the task itself and the quality of the dataset refrains me from giving a score of 5.

---

> > > ### Author Response · Authors · 2025-08-05
> > > **Response to Reviewer kYPv**
> > >
> > > Thank you so much for your positive feedback and your recommendation! To further address your remaining comments, we would like to provide additional clarification on the results of the Increased Human-Robot Interaction (Increased HRI) and Human-in-the-Loop (HITL) experiments.
> > >
> > > Increased HRI: The goal of COOPERA is to enable the study of long-term, open-ended HRC, which better mirrors real-world use, unlike most HRC benchmarks focused on episodic, closed-set tasks. We hope to encourage future research on building robot agents that can work over long time horizons and adapt to human preference. COOPERA currently focuses on day-level adaptation, which we believe better reflects real-world use—where the robot is optimized overnight while the human rests. Our additional Increased HRI experiment simulates bidirectional, hourly communication between the human and the robot, as a step toward more interactive learning. However, this remains a challenging problem: ideally, it would require an online learning algorithm (i.e., beyond just prompting/in-context learning) that enables both the human and the robot to adapt to each other simultaneously. This essentially becomes a multi-agent collaboration problem, which is beyond the scope of this work. Our designed benchmark and method currently target day-level optimization of an assistive robot.
> > >
> > > HITL: After discussing with the users, we attribute the slight performance improvement to their tendency to recall and describe mid-day activities during robot collaboration. Due to time constraints, we recruited three users mid-week, most of who are students willing to help, so their mid-day activities were relatively consistent. We believe this also demonstrates the diversity and variability modeled by our simulated humans, as they can potentially be more diverse than real humans.
> > >
> > > That being said, we will integrate both experiments into the paper with more detailed explanations. For HITL, we will expand our study with a broader participant pool to increase population diversity and further evaluate real human-robot collaboration.
> > >
> > > We are more than happy to elaborate on any further points. Thank you again for your time, thoughtful questions, and helpful discussion, which have greatly helped improve the quality of our paper!

---

### Author Response · Authors · 2025-08-08
**Author Summary of Rebuttal and Discussion Period and Thank Note for Reviewers and AC**

Dear All,

Thank you to everyone for the time and active engagement during the rebuttal and discussion period! Because of your responsible, insightful, and kind contributions, our rebuttal page has grown long. We would therefore like to provide a short summary for the convenience of all reviewers and the AC.

Before the rebuttal, the novelty and contributions of COOPERA were kindly recognized by all reviewers, with a shared understanding of its clear motivation to enable and study continual, open-ended human–robot collaboration. Specific aspects of COOPERA were also appreciated by individual reviewers. For example, the design and innovation of our human simulation pipeline (Reviewers kYPv, irvc), the thoughtfulness and comprehensiveness of our benchmark design (Reviewers Zf7z, irvc), and the breadth of our experiments (Reviewers Zf7z, irvc, PjNb).

At the same time, reviewers raised valuable questions and suggestions, which we carefully addressed through additional experiments and clarifications. After the engaged and constructive discussion, all reviewers provided thoughtful and positive feedback, which we will integrate into the next version of the paper.

After the discussion period, we are grateful to have received a mix of Accepts (5) and Borderline Accepts (4). We sincerely thank all reviewers and the AC once again for the time, effort, and excellent suggestions that have helped strengthen our work.

---

### Note · Authors · 2025-08-14

Dear Reviewers and Chairs,

We want to thank you again for your time and engagement throughout this review process. Your insights have greatly strengthened our work!

As we wrap up, we have prepared a summary of our rebuttal and discussion process. Please refer to our final comment titled "Author Summary of Rebuttal and Discussion Period and Thank Note for Reviewers and AC" for details and the positive outcomes of our discussion.

Thank you again for your support throughout this period!

---

### Decision · Program_Chairs · 2025-09-17

**Decision:**

Accept (spotlight)

**Comment:**

This paper introduces COOPERA, an Human-Robot Collaboration framework designed to facilitate open-ended collaboration with humans over long horizons, specifically considering the traits demonstrated by (simulated) humans.  The reviewers appreciated both the framework and the simulation of human behavior.  Ultimately, all reviewers agree that the paper should be accepted — the authors’ rebuttal caused most of the reviewers to raise their scores.